# Quantics Tensor Train for solving Gross-Pitaevskii equation

Aleix Bou-Comas[1★], Marcin Płodzień[2,3] Luca Tagliacozzo[1] and Juan José García-Ripoll[1]

**1** Institute of Fundamental Physics IFF-CSIC, Calle Serrano 113b, Madrid 28006, Spain
**2** ICFO-Institut de Ciencies Fotoniques, The Barcelona Institute of Science and Technology, 08860 Castelldefels (Barcelona), Spain
**3** Qilimanjaro Quantum Tech, Carrer de Veneçuela 74, 08019 Barcelona, Spain

★ aleix.bou@iff.csic.es

## Abstract

We present a quantum-inspired solver for the one-dimensional Gross–Pitaevskii equation in the Quantics Tensor-Train (QTT) representation. By evolving the system entirely within a low-rank tensor manifold, the method sidesteps the memory and runtime barriers that limit conventional finite-difference and spectral schemes. Two complementary algorithms are developed: an imaginary-time projector that drives the condensate toward its variational ground state and a rank-adapted fourth-order Runge–Kutta integrator for real-time dynamics. The framework captures a broad range of physical scenarios—including barrier-confined condensates, quasi-random potentials, long-range dipolar interactions, and multicomponent spinor dynamics—without leaving the compressed representation. Relative to standard discretizations, the QTT approach achieves an exponential reduction in computational resources while retaining quantitative accuracy, thereby extending the practicable regime of Gross–Pitaevskii simulations on classical hardware. These results position tensor networks as a practical bridge between high-performance classical computing and prospective quantum hardware for the numerical treatment of nonlinear Schrödinger-type partial differential equations.

# 1    Introduction

Since the dawn of modern science, the language of nature has been cast in the form of differential equations. From Newton's principia to Leibniz's formalism, the dynamical laws of mechanics, electromagnetism, and later, quantum theory have all taken shape as partial differential equations (PDEs). Whether Maxwell's equations for light, Navier–Stokes for fluids, or the Schrödinger equation for quantum matter, these PDEs distill physical intuition into a compact set of local rules. By encoding local conservation laws and symmetries, PDEs give a concise mathematical model of physical phenomena and remain the backbone of theoretical exploration across physics, chemistry, and engineering, in a form that bridges theory and experiment.

The digital computer revolution of the 20th century allowed the study of PDEs that do not have closed-form solutions. Finite-difference, finite-element, and spectral methods now let us march a discretized equation step-by-step in (simulation) time. Nevertheless, high-fidelity simulations of strongly nonlinear or high-dimensional systems still face an exponential wall: memory and runtime scale poorly with system size or required resolution. This "curse of dimensionality" motivates the search for more compact representations of PDEs solutions.

Over the recent years, quantum computing has been heralded as a future route, promising dramatic speed-ups for linear algebra kernels intrinsic to PDE solvers. Early work demonstrated that matrices arising from finite-element discretizations can be block-encoded and solved in polylogarithmic time, laying the foundation for quantum finite-element solvers [1,2]. Spectral formulations subsequently showed that global basis sets permit even sharper precision scaling [3]. Moving beyond linear systems, differentiable-circuit ansätze and kernel-based regressors have been proposed for nonlinear differential equations [4,5], and effective-Hamiltonian strategies now offer a physics-informed route to general nonlinear PDEs [6,7]. Hybrid demonstrations on near-term hardware already hint at practical speedups for modest grid sizes [8,9]. However, currently, a scalable and fault-tolerant quantum computer to solve PDEs does not exist [10].

An alternative to quantum computing for PDE is to exploit ideas borrowed from classical simulation of quantum many-body physics. Tensor networks such as matrix-product states (MPS) and tree tensor networks capture only the physically relevant corner of an otherwise enormous Hilbert space, taming the "curse of dimensionality" on today's classical hardware. Adapting this machinery to field equations like the nonlinear Schrödinger equation (NLSE) reveals a fertile middle ground—quantum-inspired algorithms that shrink computational costs without waiting for quantum processors.

Tensor networks were originally introduced to efficiently represent tensors with exponentially many elements by expressing them as contractions of networks of smaller tensors. Each small tensor captures a correlated subset of the full data, enabling a compact and structured encoding of complex, high-dimensional information [11–20]. Developed initially in the context of many-body quantum physics, tensor networks were designed to simulate strongly correlated quantum systems and quantum computations. Over time, however, their utility has proven to extend far beyond quantum mechanics.

For instance, MPSs were proposed for solving differential equations [21] or, compressing images [22]. Connections between tensor networks and wavelet-based signal processing have been proposed through Multi-scale Entanglement Renormalization ansatz (MERA) [23, 24].

In parallel, the mathematical community independently rediscovered MPS under the name *Tensor Trains* (TT), and developed related approximation algorithms such as the TT-cross method. TT decomposition established that exponentially large state vectors and operators admit compact, low-rank factorizations. The TT and quantized-TT (QTT) constructions, together with explicit low-rank forms of canonical operators such as the Laplacian and time-propagators, yielded rigorous error bounds and fast linear-algebra primitives that turned curse-of-dimensionality problems into tractable ones [25–29]. These structural insights have since been elevated to fully fledged, high-dimensional PDE solvers. Reformulations based on backward stochastic differential equations further extend the reach of TT methods to long-time parabolic evolution [30–33].

Application-driven studies demonstrate that the same compression principles translate into concrete physics and engineering tasks, including TT solvers for neutron transport [34], quantum-inspired computational fluid dynamics [35–37], reduced-order turbulence modeling [38], many-body Schrödinger dynamics [39], and plasma collisions [40]; all report order-of-magnitude savings in memory and runtime while maintaining controllable accuracy. These efforts led to a convergence of ideas and growing cross-fertilization between quantum physicists and numerical analysts. This emerging synergy has given rise to a new field: *quantum-inspired computational techniques* [29, 41–43].

In this work, we focus on the Gross–Pitaevskii (GPE) equation—a nonlinear Schrödinger equation describing dilute Bose gases and widely used to model Bose–Einstein condensates (BECs). Despite being relatively tractable in low dimensions, the GP equation serves as an ideal testbed for illustrating the capabilities and versatility of our approach. We extend the framework of quantum-inspired methods to handle nonlinearities and long-range potentials, a crucial step that allows us to tackle nonlinear differential equations. Furthermore, we introduce a new formalism to represent and evolve multiple species of BECs in a single MPS.

The manuscript is organized as follows: In Section 2 we recall the derivation of the Gross-Pitaevskii equation starting from the BEC description in the second quantization. Next, in Section 3 we introduce Quantics Tensor Train (QTT) formalism. In Section 4 we extend QTT to treat non-linearities, long-range interactions, and multiple BEC species with tensor trains. In Section 5 we introduce imaginary-, and real-time evolution algorithms to calculate the ground state of a system and the time-evolution of a given initial state. In Section 6 we apply our methodology to a set of physical examples including ground state calculations in a harmonic trap, the evolution of the wavepacket in quasi-disordered potential, the evolution of

105  wavepacket with long-range interactions, and spinor-BEC dynamics. We discuss the numerical
106  resources in Section 7, and conclude in Section 8.

## 2  Preliminaries: Gross-Pitaevskii equation

108  In this section, we will briefly recap the derivation of GPE in the mean-field description of BECs
109  [44]. BEC is a dilute gas of bosons cooled to temperatures so low that a macroscopic number
110  of particles occupy the same quantum state. The GPE—mathematically a cubic nonlinear
111  Schrödinger equation—provides a remarkably accurate mean–field description of a weakly–
112  interacting BEC.

113      Let us consider $N$ bosons of mass $m$ moving in an external potential $\hat{V}(\mathbf{x})$ and interacting
114  via a translationally invariant two–body potential $\hat{U}(\mathbf{x}, \mathbf{y}) = \hat{U}(|\mathbf{x} - \mathbf{y}|)$. The second–quantised
115  Hamiltonian reads

$$
\begin{aligned}
\hat{H} = &\int d^3\mathbf{x}\, \hat{\Psi}^\dagger(\mathbf{x}) \hat{H}_0 \hat{\Psi}(\mathbf{x}) \\
&+ \tfrac{1}{2} \int d^3\mathbf{x}\, d^3\mathbf{y}\, \hat{\Psi}^\dagger(\mathbf{x}) \hat{\Psi}^\dagger(\mathbf{y}) \hat{U}(\mathbf{x}, \mathbf{y}) \hat{\Psi}(\mathbf{y}) \hat{\Psi}(\mathbf{x}),
\end{aligned}
\tag{1}
$$

116  where $\hat{H}_0 = -\frac{\hbar^2}{2m}\nabla^2 + V(\mathbf{x})$ is a single-body Hamiltonian describing non-interacting bosons in a
117  trapping potential $\hat{V}(\mathbf{x})$, usually considered as a harmonic trap $\hat{V}(\mathbf{x}) = \frac{m}{2}(\omega_x^2 x^2 + \omega_y^2 y^2 + \omega_z^2 z^2)$.
118  The bosonic field operators satisfy the canonical equal–time commutator $[\hat{\Psi}(\mathbf{x}), \hat{\Psi}^\dagger(\mathbf{y})] = \delta^{(3)}(\mathbf{x} - \mathbf{y})$,
119  $[\hat{\Psi}, \hat{\Psi}] = [\hat{\Psi}^\dagger, \hat{\Psi}^\dagger] = 0$, and the particle–number operator $\hat{N} = \int d^3\mathbf{x}\, \hat{\Psi}^\dagger(\mathbf{x}) \hat{\Psi}(\mathbf{x})$ commutes
120  with $\hat{H}$, ensuring number conservation. Next, let us expand field operators in any orthonor-
121  mal single–particle basis $\{\varphi_i\}$ (e.g. eigenfunctions of the $\hat{H}_0$), $\hat{\Psi}(\mathbf{x}, t) = \sum_{i=0}^\infty \hat{a}_i(t) \varphi_i(\mathbf{x})$, with
122  bosonic ladder operators $[\hat{a}_i, \hat{a}_j^\dagger] = \delta_{ij}$. Now, let us consider a system below the critical tem-
123  perature $T_c$. For $T < T_c$, a single mode (taken to be $i = 0$) acquires macroscopic occupation
124  $\langle \hat{a}_0^\dagger \hat{a}_0 \rangle \sim \mathcal{O}(N)$. Splitting off fluctuations $\hat{a}_0(t) = e^{-i\theta(t)}\big(\sqrt{N} + \hat{\delta}_0(t)\big)$, $\langle \hat{\delta}_0 \rangle \ll \sqrt{N}$ then ne-
125  glecting $\hat{\delta}_0$ and all non–condensed modes $\hat{a}_{i\neq 0}$ gives the $c$–number substitution $\hat{\Psi}(\mathbf{x}, t) \longrightarrow \sqrt{N}\, \varphi(\mathbf{x}, t)$,
126  $\varphi(\mathbf{x}, t) \equiv e^{-i\theta(t)}\psi(\mathbf{x})$, with normalisation $\int d^3\mathbf{x}\, |\psi|^2 = 1$. Substituting the classical field into
127  Eq. (1) and using the bosonic commutation relations yields an ordinary functional of the $c$–
128  number field:

$$
\begin{aligned}
\mathcal{E}[\psi, \psi^*] = &\, N \int d^3\mathbf{x}\, \psi^*(\mathbf{x}) \hat{H}_0 \psi(\mathbf{x}) \\
&+ \frac{N^2}{2} \int d^3\mathbf{x}\, d^3\mathbf{y}\, |\psi(\mathbf{x})|^2 \hat{U}(|\mathbf{x} - \mathbf{y}|) |\psi(\mathbf{y})|^2.
\end{aligned}
\tag{2}
$$

129      A stationary condensate is described by a time–independent $\psi(\mathbf{x})$ that minimises the en-
130  ergy while preserving particle number. Introducing a Lagrange multiplier $\mu$ (chemical po-
131  tential), defining functional $\mathcal{F}[\psi, \psi^*] = \mathcal{E}[\psi, \psi^*] - \mu N \int |\psi|^2$, and varying $\psi$, one finds the
132  time-independent GPE

$$
\left[ \hat{H}_0 + N \int d^3\mathbf{y}\, \hat{U}(|\mathbf{x} - \mathbf{y}|) |\psi(\mathbf{y})|^2 \right] \psi(\mathbf{x}) = \mu\, \psi(\mathbf{x}).
\tag{3}
$$

133      The time-evolution equation for the mean-field function can be obtained by extremising a
134  real-time action that reproduces the correct Hamiltonian equations of motion:

$$
S[\psi, \psi^*] = \int dt\, d^3\mathbf{x} \left[ \frac{i\hbar N}{2}\big(\psi^* \partial_t \psi - \psi \partial_t \psi^*\big) - \mathcal{E}[\psi, \psi^*] \right],
\tag{4}
$$

where $\mathcal{E}$ is the energy density appearing in Eq. (2). The Euler–Lagrange equation yields

$$i\hbar\partial_t\psi(\mathbf{x},t) = \left[\hat{H}_0 + N\hat{U}_{\text{int}}(\mathbf{x})\right]\psi(\mathbf{x},t),$$
$$\hat{U}_{\text{int}}(\mathbf{x}) = \int d^3\mathbf{y}\,\hat{U}(|\mathbf{x}-\mathbf{y}|)|\psi(\mathbf{y},t)|^2 \tag{5}$$

Finally, for a dilute gas, the two-body potential can be replaced by the pseudopotential $\hat{U}(\mathbf{r}) = g\,\delta^{(3)}(\mathbf{r})$, where the coupling strength reads $g = \frac{4\pi\hbar^2 a_s}{m}$, and $a_s$ is the $s$-wave scattering length. Then the spatial integral in Eqs. (5) simplifies, giving the Gross-Pitaevski (GP) equation

$$i\hbar\partial_t\psi(\mathbf{x},t) = \left[\hat{H}_0 + gN|\psi(\mathbf{x},t)|^2\right]\psi(\mathbf{x},t), \tag{6}$$

The GPE is the basis of most of our current understanding of the physics of BECs, and it accurately describes any experimental setups of cold bosonic systems, which interact weakly. The GPE can be generalized to describe different physical scenarios. For example, it can describe mixtures of different atomic species or the physics of charged BEC which requires considering long-range potentials. All these scenarios can be described by a generalized form of the GPE encoded by systems of partial differential equations for $M$ BEC species,

$$i\hbar\partial t\psi_i(\boldsymbol{x}) = \left(\hat{H}_0 + \sum_{j=1}^{M} g_{ij}\sqrt{N_i N_j}|\psi_j(\boldsymbol{x})|^2\right)\psi_i(\boldsymbol{x})$$
$$+ \sum_{j\neq i}\lambda_{i,j}\psi_j(\boldsymbol{x}) \tag{7}$$

where $g_{ij}$ is an interaction matrix encoding the interaction among the different BEC species with $N_i$ particles each, and the interaction potential can contain both local $U(\mathbf{x})$, and the long-range part $U_{LR}(|\mathbf{x}-\mathbf{x}'|)$. The two-body long-range interactions between BEC atoms appear in a mean-field description as a convolution with the BEC density, i.e.

$$V_{LR}(\boldsymbol{x}) = \int d\boldsymbol{x}'|\psi(\boldsymbol{x}')|^2 W\left(|\boldsymbol{x}-\boldsymbol{x}'|\right). \tag{8}$$

Finally, we shortly discuss the experimental control over the dimensionality of BEC. When a harmonic trap is highly anisotropic $\omega_\perp \equiv \omega_y = \omega_z \gg \omega_x \equiv \omega_0$, then the dynamics can be effectively reduced to one direction. Writing $\psi(\mathbf{x}) = \psi(x)\varphi_\perp(y,z)$ with the transverse ground state $\varphi_\perp$ assumed to be the Gaussian state with width $a_\perp$, and integrating over the frozen coordinates, gives an effective one-dimensional GPE (1D-GPE)

$$i\hbar\partial_t\psi(x,t) = \left[\hat{H}_0 + g_{1D}N|\psi|^2\right]\psi(x,t), \tag{9}$$

where $g_{1D} = \frac{g}{2\pi a_\perp^2}$, and $\hat{H}_0 = -\frac{\hbar^2}{2m}\partial_x^2 + V_{1D}(x)$. Similarly, for a pancake trap, $\psi(\mathbf{x}) = \psi(x,y)\phi(z)$, $\omega_\perp \gg \omega_x = \omega_y \equiv \omega_0$, one obtains $g_{2D} = g/\sqrt{2\pi}a_z$.

In the following, we restrict our studies to quasi-one dimensional geometries choosing $l_0 = \sqrt{\hbar/m\omega_0}$, $t_0 = 1/\omega_0$, $E_0 = \hbar\omega_0$ as length, time, and energy units, respectively.

## 3  Tensor Networks Formalism for calculus

A tensor network is a decomposition of a high-rank tensor into a contraction of smaller set of constituent tensors. The simplest tensor networks, called matrix product states (MPS) or

tensor trains (TT), consist in a sequential contraction of two- or three-legged tensors. They were discovered as an efficient representation of stronly-correlated one-dimensional many-body systems and Markov chains [45, 46] and later related to the solutions of the DMRG algorith [47, 48].

In recent years, the study of MPS has evolved beyond quantum physics and it has been shown to efficiently compress a wide set of analytical one-dimensional functions, [41, 42, 49–52]. The basic idea is that a real function of one real variable $f(x): \mathbb{R} \to \mathbb{R}$ can be stored in a computer only after appropriately defining a one-dimensional lattice $\Lambda$ of $N_{\text{grid}}$ points $x_i \in \Lambda$ (we store $T_{x_i} = f(\{x_i\}_{i \in \mathcal{I}})$). Here the idea is that the representation of the function will be better as we choose finer and finer lattices, and thus as $N \to \infty$.

Similarly, a function $f(x, y): \mathbb{R}^2 \to \mathbb{R}$ of two real variables defined on a bounded region can be encoded as a finite matrix once the two variables are discretized. This means that given two one-dimensional lattices $\Lambda_x$ and $\Lambda_y$ each with $N_{\text{grid}}$ points and $x_i \in \Lambda_x$ and $y_j \in \Lambda_y$ we can define the $N_{\text{grid}} \times N_{\text{grid}}$ matrix of real numbers containing its restriction on the lattice points as

$$F(i, j) \equiv f(x_i, y_j) \in R, \tag{10}$$

A function of a single variable can become a tensor with high rank, by discretizing (or quantizing) the continuous variable in a specific basis [22]. For simplicity, let's consider we chose 2 as our basis and we rescale the domain of the function $f$ such that $\bar{x} \in [0, 1]$. We can express $\bar{x} = \sum_{j=0}^{\infty} b_j 2^{-j}$ and we obtain a tensorized form of our function as $T_{\bar{x}} = T_{b_1,\dots,b_\infty}$. Notice that as expected a function of a continuous variable using this strategy is encoded by a tensor with infinitely many indices. If we decide to discretize the interval $[0, 1)$ on a lattice with lattice spacing $2^{-n}$, $\Lambda_{\bar{x}}$ made by points $N_{\text{grid}}$ $\bar{x}_i$ we obtain a finite rank tensor, $T(x_i) = T_{b_1,\dots,b_n}$ with only $n = \log_2(N_{\text{grid}})$ indices. This means that the number of points we consider scales exponentially with the number of indices of the tensor.

The discretization of a function as a tensor with $n$ indices may be reinterpreted as the wavefunction of a quantum many-body system of $n$ spin-1/2 constituents. This many-body state, regarded as a wavefunction, may be compressed using different tensor-networks techniques, depending on the entanglement structure. The simplest choice is to compress the function using a one-dimensional MPS.

Given a quantum many-body system of $n$ constituents $|\psi\rangle$, each described by an individual Hilbert space spanned by vectors $|s_i\rangle$, the MPS representation of the system compresses the wavefunction of it as the contraction of $n$ three leg tensors $\phi_{s_j}^{r_{j-1}, r_j}$, one for each constituent. Each three-leg tensor has one index ($s_j$) encoding the state of the constituent and two extra indices, $r_{j-1}, r_j$ which allows us to contract the tensor with its left and right neighbors following the desired one-dimensional pattern.

$$|\psi\rangle = \sum_{s_1, s_2, \dots, s_n} c_{s_1, s_2, \dots, s_n} |s_1 s_2 \dots s_n\rangle$$
$$\approx \phi_{s_1}^{r_1} \phi_{s_2}^{r_1, r_2} \phi_{s_3}^{r_2, r_3} \dots \phi_{s_n}^{r_{n-1}} |s_1 s_2 \dots s_n\rangle \tag{11}$$

where $s_i$ are the physical degrees of freedom while $r_i$ are contracted indices between adjacent tensors. Notice that in the second line of the formula above we have omitted the sum over repeated indices, using the Einstein notation.

Using the same construction, the function $T(x_i) = T_{b_1,\dots,b_n}$ can be encoded as

$$T(x) = T_{b_1, b_2, \dots, b_n} \approx \phi_{b_1}^{r_1} \phi_{b_2}^{r_1, r_2} \phi_{b_3}^{r_2, r_3} \dots \phi_{b_n}^{r_{n-1}} \tag{12}$$

in this case, we have replaced the physical degrees of freedom $s_i$ for $b_j$ which represents the bits and are the external legs, while $r_i$ are the links between tensors, their dimensions are called bond dimensions $\chi_i = \dim(r_i)$. The specific case of MPS we are discussing here, where

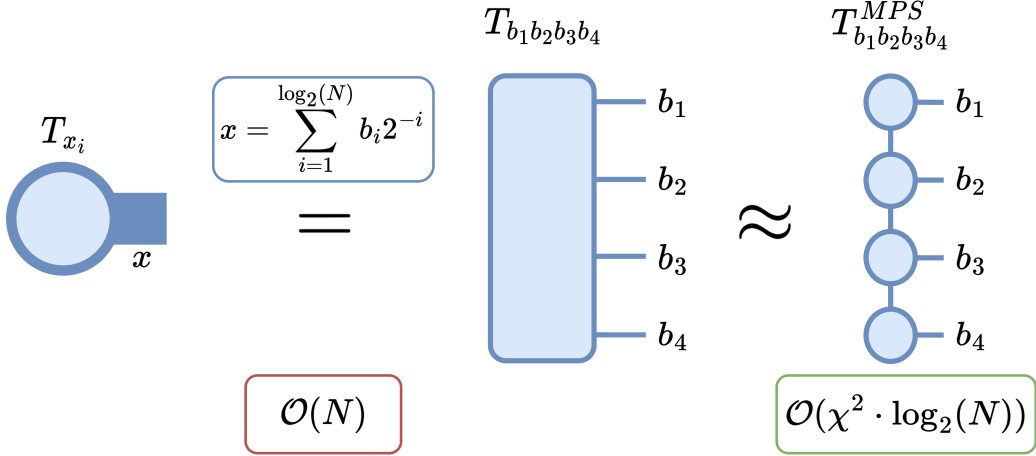

Figure 1: Three possible representations of how to store a vector in a computer. From left to right we have a vector, a tensor, and a Matrix Product State (MPS). The first and the second representations are exact and the only difference between them is the basis of $i \in \mathcal{I}$ chosen: in the first case $\mathcal{I} = \{1, 2, \ldots, N\}$ and in the second case, $i$ is expressed through the binary representation. The third method follows a controlled truncation method to yield a faithful representation of the previous tensor requiring a smaller number of parameters.

$x$ is discretized using a binary encoding to compress a function is also called Quantics Tensor Train (QTT). With an abuse of notation, we will refer to the QTT representation of $f(x)$ as $T(x) = \langle x|f \rangle$. A representation of such encoding is depicted in Fig 1. Notice that while the $r$ indices are summed over, differently from the wave-function case, we do not have to sum over the $\{b_l\}$ indices, since each choice of the $b_i$ represents a different point of grid $x_i$.

The performance of the MPS formalism depends on how large the elementary tensors need to be to obtain a faithful representation of the initial tensor. Assuming that all the vector spaces labeled by $r$ have dimension $\chi$, the above expression compresses a vector of $N_{\text{grid}}$ parameters as a function of the entries of the elementary tensors which are of the order $\mathcal{O}\left(2\chi^2 \log_2(N_{\text{grid}})\right)$[1]. Therefore, the MPS representation is efficient as long as it requires elementary tensors whose bond dimension does not depend on $n$ or increases sub-exponentially with it.

In one dimension, there are a large number of functions in which $\chi$ stays constant while the number of points increases [49], therefore the QTT representation is efficient in compressing those functions. Furthermore, there are proofs that regular enough functions can be efficiently compressed as TT with fixed bond dimension [49, 51, 52]. Beyond one dimension there is still no formal proof of such a statement, but the benchmark calculations performed so far point towards similar results [40, 50, 53].

The inner product of two QTT, $|f\rangle$ and $|g\rangle$, is the addition of all the values $f(x_i)g(x_i)$, thus, with the correct measure it can be interpreted as the integral of the product of the functions,

$$\langle g|f \rangle \Delta x = \sum_{i=1}^{N} g(x_i)f(x_i)\Delta x \simeq \int_0^1 \mathrm{d}x\, g(x)f(x) \tag{13}$$

where $\Delta x$ is the discretization size. The convergence of the integral can be faster using different quadratures [54], but it is always exponential in the number of qubits.

---

[1]Given that the computational cost increases with the bond dimension, we can estimate an upper bound of it by assuming that all the tensor share the largest bond dimension which we refer to as $\chi = \max_i(\chi_i)$.

²²⁴     The QTT representation can be generalized to compress operators acting on functions. In
²²⁵ quantum mechanics, we can compress an operator $U$ acting on a collection of spins $\{s_1, \ldots, s_n\}$

$$
\begin{aligned}
U &= \sum_{\{s\},\{s'\}} c_{s_1,\ldots,s_n}^{s'_1,\ldots,s'_n} |\{s_i\}\rangle \langle\{s'_i\}| \\
&\approx \sum_{\{s\},\{s'\}} \phi_{s_1,s'_1}^{r_1} \phi_{s_2,s'_2}^{r_1,r_2} \ldots \phi_{s_n,s'_n}^{r_{n-1}} |\{s_i\}\rangle \langle\{s'_i\}|
\end{aligned}
\tag{14}
$$

²²⁶ where, $|\{s_i\}\rangle = |\{s_1, \ldots, s_n\}\rangle$, $|\{s'_i\}\rangle = |\{s'_1, \ldots, s'_n\}\rangle$, and as before, $r_i$ are the bonds that connect
²²⁷ the four-legged tensors and $s_i$ are the degrees of freedom. This object is called a Matrix Product
²²⁸ Operator (MPO).
²²⁹     Similarly, operators that act onto functions of continuous variables, once these have been
²³⁰ appropriately discretized, are represented by finite dimensional matrices. Such matrices can
²³¹ be represented through MPO,

$$
\mathcal{T}(f'(x), f(x)) = \mathcal{T}_{b_1,b_2,\ldots,b_n}^{b'_1,b'_2,\ldots,b'_n} \approx \phi_{b_1,b'_1}^{r_1} \phi_{b_2,b'_2}^{r_1,r_2} \ldots \phi_{b_n,b'_n}^{r_{n-1}}.
\tag{15}
$$

²³² The MPOs describe the $\mathcal{O}\left(N_{\text{grid}}^2\right)$ matrix elements, while compressing them only using $\mathcal{O}\left(4k^2 \log_2(N_{\text{grid}})\right)$
²³³ independent parameters, where $k$ is the maximum bond dimension of the MPO, $k = \max_i(\dim(r_i))$.
²³⁴ If $k$ increases mildly (at most polynomially) with the number of bits ($\log_2(N_{\text{grid}})$) such a com-
²³⁵ pression is advantageous.
²³⁶     Besides the compression of functions and operators, tensor networks also allow to stream-
²³⁷ line their manipulation. For example, the matrix-vector multiplication requires $\mathcal{O}\left(N_{\text{grid}}^2\right)$ op-
²³⁸ erations, but the number of operations required to perform the MPS-MPO multiplication is
²³⁹ upper-bounded by $\mathcal{O}\left(4\log_2(N_{\text{grid}})\left(\chi^3 k + \chi^2 k^2\right)\right)$.
²⁴⁰     In this work, we will mostly concentrate on solving partial differential equations (PDEs)
²⁴¹ with QTT. This approach is motivated partially by the observation that most of the MPO en-
²⁴² coding of the operators used in PDEs requires a small bond dimension. For example, the
²⁴³ construction of the first derivative of a function, expressed using first-order finite differences
²⁴⁴ requires $\chi = 3$, and the second derivative requires $\chi = 5$ as explained e.g. in [40].

## 3.1  Loading a function into QTT

²⁴⁶ The first challenge found when working with QTTs is to efficiently compute a generic functions'
²⁴⁷ tensor representation. We could of course interrogate the values of a function over all $\mathcal{O}(N_{\text{grid}})$
²⁴⁸ points on the grid, using iterative singular value decompositions to obtain all tensors would
²⁴⁹ cost $\mathcal{O}(N_{\text{grid}}^{3/2})$. However, that would eliminate any advantage of working with tensor networks
²⁵⁰ and would restrict our computations to grids where $N_{\text{grid}}$ is a number that fits in the computer's
²⁵¹ memory. There are three solutions to this problem: (i) a direct estimation of the QTT tensors,
²⁵² (ii) the use of polynomial interpolation, and (iii) a clever and adaptive sampling of the function
²⁵³ via Tensor Cross Interpolation (TCI).
²⁵⁴     A few functions can be directly encoded as quantized tensor trains. These include (i)
²⁵⁵ exponentials, (ii) sums of functions, (iii) sinusoidal functions and (iv) polynomials. First,
²⁵⁶ let us take the discretization of $f(x) = \exp(\alpha x)$ over a grid with $2^n$ points. By writing the
²⁵⁷ $x \in [a, b)$ coordinate in binary notation $x_i = a + (b-a)\sum_{j=1}^{n} b_j 2^{-j}$, the exponential becomes
²⁵⁸ a product of rank-1 tensors

$$
f(x_i) = \exp\left(\alpha \sum_{j=1}^{n} b_j 2^{-j}\right) = \prod_{j=1}^{n} \exp\left(\alpha b_j 2^{-j}\right).
\tag{16}
$$

259  Each tensor $\phi_{b_j}^{r_{j-1},r_j}$ in Eq. (12) has $\dim(r_j) = 1$, $\forall j$, and

$$
\phi_{b_j} = |b_j = 0\rangle + \exp(\alpha 2^{-j})|b_j = 1\rangle = \begin{bmatrix} 1 \\ \exp(\alpha 2^{-j}) \end{bmatrix} \tag{17}
$$

260  involves an small number of parameters $\sim 2 \log_2(N_{\text{grid}})$.

261      Next, one may find that the sum of two functions $f(x) + g(x)$ that are encoded with tensors
262  of dimensions $\chi_f$ and $\chi_g$ can be encoded using tensors of size $\chi_f + \chi_g$, which are direct sums
263  of the original QTT tensors. This explains why sinusoidal functions, $\sin(x) = \frac{\exp(ix) - \exp(-ix)}{2i}$
264  and $\cos(x)$, require tensors of dimension $\chi = 2$.

265      As explained in Refs. [42, 51, 52] there exist other algorithms to encode functions that are
266  1D-polynomials of degree $d$, using tensors of dimension $\chi = d + 1$ or less. This may be used
267  for constructing interpolating functions, using methods such as *Chebyshev* or *Lagrange* inter-
268  polation [50], using $\mathcal{O}(\log_2(N_{\text{grid}}))$ function evaluations. However, a simple generalization of
269  these ideas enables also the composition of functions $f(g(x))$, this idea is developed in [42].
270  This, in combination with the addition and multiplication of functions (c.f. Sect. 4.1) allows
271  the encoding of a large family of models that have analytical representations.

272      Having said this, polynomial and algebraic approximations are not always efficient and
273  may lead to errors in functions with large bandwidth or low differentiability properties. A
274  more general family of algorithms has emerged under the name *Tensor Cross Interpolation*.
275  These are adaptive, high-quality low-rank approximations of high-rank tensors and discretized
276  functions that require a logarithmic number of function evaluations in many cases. These
277  methods are generalizations of the *cross interpolation*, an algorithm that creates a low-rank
278  approximation of a matrix by querying select rows and columns, called pivots. Similarly, TCI
279  queries the values of a multidimensional tensor (or function) at a set progressively adapted
280  points to create a small-rank TT/MPS approximation. More information about the algorithm
281  and their generalization to tree tensor networks can be found in [55–57], while a comparison
282  of the performance of different function loading techniques is available in Ref. [42].

## 3.2  Differential operators

284  To solve partial differential equations we need to represent differential operators as matrices
285  (or tensors). One possibility is through finite differences. The matrical form of finite differ-
286  ences corresponds to sparse matrices with non-zero elements near the diagonal. The non-zero
287  elements just perform the translations of $f(x)$ that the finite difference algorithm requires.
288  For example, computing the first derivative with a central finite difference first order approx-
289  imation implies defining $f'(x) = \frac{f(x+h) - f(x-h)}{2h}$. Therefore, given $f(x)$ as QTT we need to
290  construct $f(x + h)$ and $f(x - h)$.

291      We need to create an MPO that represents the $\delta$-function $\mathcal{T}_h = \delta(y - (x + h))$ because the
292  MPO-MPS multiplication performs an integral over the shared degrees of freedom,

$$
f(x + h) = \int \mathrm{d}y f(y) \delta(y - (x + h)). \tag{18}
$$

293  This operator can be created as an MPO exactly following the *ripple-carry adder* algorithm. The
294  details of the translation matrix product state, $\mathcal{T}_a$ are explained in Appendix A. The creation
295  of the translation MPO allows to define differential operators as MPOs, for example,

$$
\begin{aligned}
\frac{\partial}{\partial x} &= \frac{\mathcal{T}_h - \mathcal{T}_{-h}}{2h} \\
\frac{\partial^2}{\partial x^2} &= \frac{\mathcal{T}_h - 2\mathbb{I} + \mathcal{T}_{-h}}{h^2}
\end{aligned} \tag{19}
$$

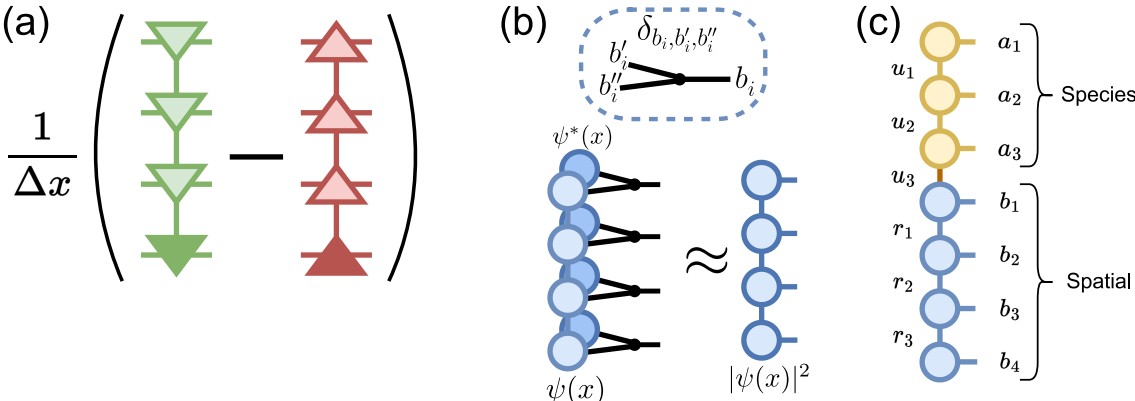

Figure 2: Illustration of different formalism explained in section 3 and 4. Figure 2(a) represents the creation of the operator $\partial_x$ with finite differences. The triangles pointing up (down) represent the positive (negative) translation of $\Delta x$, more details on how to create these tensors can be found in Appendix A. Figure 2(b) depicts the copy tensor described in section 3.3. It also shows how to create the tensor $|\psi(x)|^2$ from the individual tensors $\psi(x)$, $\psi^*(x)$, and the copy tensor following the prescription of section 4.1. Figure 2(c) represents how we can express different species of BEC with the QTT formalism, the three initial tensors encode eight different BEC species while the four final tensors represent the spatial discretization.

where $\mathbb{I}$ represents the identity MPO and $h$ is equal to the lattice spacing $\Delta x = L/2^n$. Fig. 2(a) shows a diagrammatic representation of the creation of the partial derivative $\partial_x$ at first-order of finite differences.

Ref. [40] presents an alternate method to encode differential operators. They construct the MPO that represents the differential operator directly through Pauli matrices using a similar strategy to the one introduced in the context of encoding the Hamiltonian of a many-body system into an MPO, [58]. A different strategy consists in creating the differential operator in the Fourier space, where it is encoded by a diagonal MPO, and then, transform it back to the real space. The fact that the momentum representation of a differential operator is just a simple polynomial, together with the low bond dimension required to transform from and to Fourier space [59], implies that the differential operator can also be expressed as an MPO with small bond dimension. Finally, we want to mention also the existence of a new method, which represents the free propagation of the particle using the HDAF, the algorithm is reported in Ref. in [29], and in the benchmarks performed there, they show it surpasses the accuracy of methods based on differential operators defined by finite differences.

## 3.3 Copy Tensor

The *copy tensor* is a fundamental element to solve PDEs with QTTs. It allows us to perform the tensor product of two QTT, contracting them with a three-leg copy tensor which encodes the generalized Kroenecker delta, $\delta_{b_i, b_i', b_i''}$:

$$\delta_{b_i, b_i', b_i''} = 0, \quad \forall b_i \neq b_i' \neq b_i'';$$
$$\delta_{b_i, b_i', b_i''} = 1, \quad \forall b_i = b_i' = b_i''. \tag{20}$$

Using such tensor, we can find the QTT representation for important elements of the GPE such as $|V(x)\psi(x)\rangle$, or $||\psi|^2\rangle$. Fig. 2(b) shows the copy tensor and how to use it to create $||\psi|^2\rangle$.

### 3.4 Local Potential

In traditional methods, local potentials are expressed through diagonal matrices or vectors. In the QTT formalism, the generalization is straightforward. First, we load the desired potential $V(x)$ with one of the methods discussed in section 3.1, then, a diagonal MPO is created using the copy tensor.

$$
\phi_{b_i}^{r_{i-1},r_i} \to \phi_{b_i,b_i'}^{r_{i-1},r_i} = \delta_{b_i,b_i',b_i''} \phi_{b_i''}^{r_{i-1},r_i}
$$
$$
= \begin{cases} \phi_{b_i}^{r_{i-1},r_i}, & b_i = b_i' \\ 0, & b_i \neq b_i' \end{cases} \tag{21}
$$

The resulting MPO has the same bond dimension of the original MPS representing $V(x)$. In general, we will want to represent $|V(x)\psi(x)\rangle$, the construction of $V(x)$ through the copy tensor allows us to realize such operation and it has a computational cost upper-bounded by $\mathcal{O}\left(4\log_2(N_{\text{grid}})(\chi^3 k + \chi^2 k^2)\right)$ where $\chi$ and $k$ are the maximum bond dimensions of $|\psi(x)\rangle$ and $|V(x)\rangle$, respectively.

## 4 Expanding the QTT toolbox

A natural question arises once we move beyond linear operators acting on functions, one is led to ask whether we can still use the QTT formalism to manipulate and solve the GPE, Eq.(5). The main difficulty that arises when trying to use QTT for solving the GP equation is that it is a non-linear PDE acting in complex-values functions.

In order to perform such generalization, we once more leverage on the fact that the TN toolbox has been designed originally in the context of quantum physics. There, complex numbers appear naturally since we deal with complex wave-function amplitudes and thus we simply need to use elementary tensors of the QTT which are complex-valued. Non-linearities are also typically encountered when computing Renyi entropies and in general, their computation requires higher complexity than the one encountered so far [60]. By using these inspirations we can thus generalize the QTT toolbox to deal with GP-like equations.

Besides adding the non-linearities to the QTT toolbox, we also discuss how to encode different BEC species within a single QTT and how to implement long-range potentials. All these new algorithms are the necessary building blocks to successfully simulate the GPE in interesting physical scenarios.

### 4.1 Non-linearity

The GPE differs from the Schrödinger equation because it has non-linearities. Solving the non-linear part of the GP equation is the part that demands more computational resources.

Using the copy tensor, we can define a QTT $||\psi|^2\rangle$ as the product of $\Phi_{b_i}^{(r_{i-1},r_{i-1}'),(r_i,r_i')}$, which are created via the elementary tensors $\phi_{b_i}^{(r_{i-1},r_i)}$ and $\phi^{*(r_{i-1}',r_i')}_{b_i}$,

$$
\Phi_{b_i}^{(r_{i-1},r_{i-1}'),(r_i,r_i')} = \delta_{b_i,b_i',b_i''} \phi_{b_i'}^{r_{i-1},r_i} \left(\phi_{b_i''}^{r_{i-1}',r_i'}\right)^*. \tag{22}
$$

Notice that we have labeled $r_i$ and $r_i'$ differently, therefore, they are not contracted, but $\Phi$ is defined as the tensor product of the $\phi$ and $\phi^*$ contracted with the generalized copy tensor. As a result,

$$
||\psi|^2\rangle = \Phi_{b_1}^{(r_1,r_1')}\Phi_{b_2}^{(r_1,r_1'),(r_2,r_2')}\ldots\Phi_{b_n}^{(r_{n-1},r_{n-1}')}. \tag{23}
$$

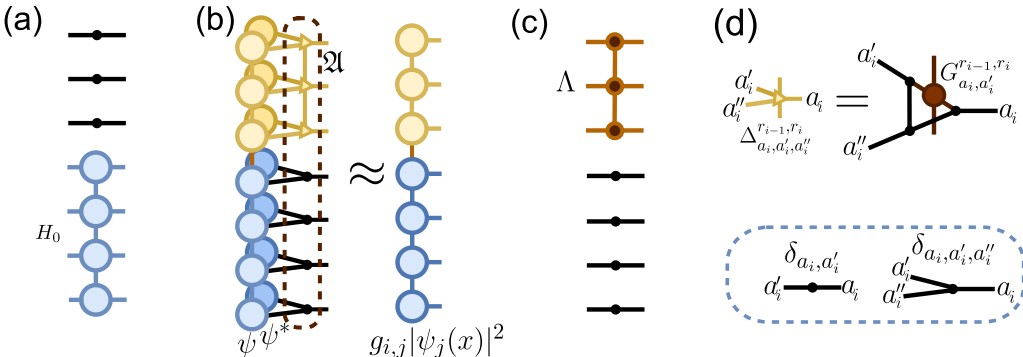

Figure 3: Illustration of different tools to simulate GPE with different BEC species described in section 4.2. The generalization of the Hamiltonian MPO for several species is described in (a). (b) shows the generalization of the copy tensor to represent the interaction term. The generalization of the copy tensor found in Eq. (28) is highlighted inside the brown dotted box. The MPO representation of the tunneling term following Eq. (29) is drawn in (c). Finally, (d) represents how one can create each generalization of the copy tensor to treat the interaction term for multiple species, it is created following Eq. (27). The dotted box acts as a legend reminding the reader that a line with a dot in the middle represents a Kroenecker delta, while three lines connected by a dot represent its generalization, the copy tensor.

Without further compression, the bond dimension of $||\psi|^2\rangle$ increases to $\chi^2$. However, we can try to further compress the state, by e.g. an iterative *QR* decomposition of the matrices, thus trying to achieve a new bond dimension $\chi'$. Our numerical experiments show that, in general, the new bond dimension $\chi' < \chi^2$.

The joint construction and truncation of $||\psi|^2\rangle$ has a computational cost upperbounded by $\sim \mathcal{O}(8\log_2(N_{\text{grid}})\chi^4)^2$. This cost is, in general, higher compared to the usual MPO-MPS multiplication involved in the solution of linear equations $\mathcal{O}(4\log_2(N_{\text{grid}})(\chi^3 k + \chi^2 k^2))$ because the bond dimension of the MPO representing the Hamiltonian is typically smaller than the bond dimension of the wavefunction, $k < \chi$. Higher non-linearities can be treated equally. For example $||\psi|^4\rangle$ can be obtained by squaring $||\psi|^2\rangle$ and thus will have a computational cost $(\chi')^4$ which is upper bounded by $\chi^8$. We thus see that with this method, the complexity scales exponentially with the degree of non-linearity. Alternative techniques might be useful, such as e.g. the function composition introduced in [42].

Since here we are dealing with the GPE that involves at most a quadratic non-linearity we will stick to the present formulation keeping in mind that if necessary, such a technique can be further improved [3].

Once we have understood how to treat the non-linear term of the GPE using Eq. (9) and its subsequent truncation we can generalize the copy tensor to include different interacting species or long-range potentials.

## 4.2  Spinor degrees of freedom

The GP equation can be generalized to model a multi-species Bose-Einstein condensate. This often requires storing $M$ vectors that encode the wavefunction of each species, $\psi^a(x)$ with

---

[2]The cost is the same to the MPO-MPS contraction $\mathcal{O}(4\log_2(N_{\text{grid}})(\chi^3 k + \chi^2 k^2))$ but in this case $\chi = k$, therefore we arribe at a computational cost $\mathcal{O}(8\log_2(N_{\text{grid}})\chi^4)$.

[3]During the preparation of this manuscript a novel TT multiplication algorithm has been proposed in [61]. It reduces the computational complexity of such multiplications from $\mathcal{O}(\chi^4)$ to $\mathcal{O}(\chi^3)$ and the memory cost from $\mathcal{O}(\chi^3)$ to $\mathcal{O}(\chi^2)$. The use of such algorithms would further reduce the computational cost of the QTT GPE solver.

$a \in \{1, \ldots, M\}$. However, we can make use of the QTT formalism that compresses analytical functions to compress the wavefunction of $M$ different species at a smaller cost. As long as there are correlations between the different species, we expect that the bond dimension of the QTT will not increase exponentially with the number of species.

Following the nomenclature of Eq. (12) the MPS would have the form of

$$T_{a,b_1,\ldots,b_n} \approx \varphi_{a_1}^{u_1} \cdots \varphi_{a_m}^{u_{m-1},u_m} \phi_{b_1}^{u_m,r_1} \cdots \phi_{b_n}^{r_{n-1}} \tag{24}$$

where we have stressed out the difference between the different species tensor and the spatial tensors calling them $\varphi$ and $\phi$ respectively. Additionally, each BEC species is labeled as $a \in \{0, M-1\}$ and then $a$ is expressed in the binary base, $a = \sum_{j=0}^{\log_2(M)-1} a_j 2^j$. Fig. 2(c) shows a graphic representation of the multi-species wavefunction encoded in a single QTT. In this case, we have 3 bits representing 8 different BEC species (yellow circles) while four spatial bits discretizing the domain into $2^4$ points (blue circles).

Eq. (7) models the behavior of the multi-species GPE. This equation corresponds to an experiment where all the species are under the same potential, the interaction between each pair of them is of the form $g_{ij}|\psi_j(x)|^2\psi_i(x)$, and they can also spontaneously tunnel from one species to another, as it indicates the tern $\lambda_{i,j}\psi_j$. Consequently, we need to generalize our construction of the Hamiltonian evolution $\hat{H}_0$ and interacting term $g|\psi|^2\psi$ to a representation that takes into account the different species. We also need to create the tunneling term. In the following paragraphs, we will detail how to construct all these elements through generalizations of the copy tensor.

We start with the simplest case, the generalization of the Hamiltonian $\hat{H}_0$ for several species. The key feature is that all the species are under the same potential, therefore assuming we can represent the single species Hamiltonian as an MPO. The Hamiltonian can be generalized into a multi-species Hamiltonian by adding Kroenecker deltas in the bits representing species,

$$\begin{aligned} \hat{H}_0^{\text{single}} &= \phi_{b_1,b_1'}^{r_1} \phi_{b_2,b_2'}^{r_1,r_2} \cdots \phi_{b_n,b_n'}^{r_{n-1}} \rightarrow \\ \hat{H}_0^{\text{multi}} &= \delta_{a_1,a_1'} \cdots \delta_{a_m,a_m'} \phi_{b_1,b_1'}^{r_1} \phi_{b_2,b_2'}^{r_1,r_2} \cdots \phi_{b_n,b_n'}^{r_{n-1}}. \end{aligned} \tag{25}$$

The Kroenecker delta forces the Hamiltonian to act on a single species, it avoids their mixture. A graphical representation of the final MPO is depicted in Fig. 3(a), where the disconnected black lines with a dot in the middle represent the Kroenecker deltas, while the connected blue circles represent the spatial MPO that encodes the Hamiltonian $H_0$.

The second case we will explain is the interaction case. In this case, we will require two different operations; in the spatial bits, we will use the copy tensor to create $|\psi_j|^2$, and in the bits that encode the different species we will use a generalization of the copy tensor to generate mixing between species. The generalization of the copy tensor performs the following action, first of all, it pairs two of the three indices to be the same (because we want to create $|\psi_j|^2$) but, the third index is free because we want to mix the different species with the weights given by the interaction tensor $g_{a_1,a_2,\ldots,a_m}^{a_1',a_2',\ldots,a_m'}$. The interaction tensor can be transformed into an MPO using brute force if the number of species is small or using TCI as mentioned in section 3.1. Assuming we have been successful in creating the interacting MPO,

$$g_{a_1,a_2,\ldots,a_m}^{a_1',a_2',\ldots,a_m'} \approx G_{a_1,a_1'}^{r_1} G_{a_2,a_2'}^{r_1,r_2} \cdots G_{a_m,a_m'}^{r_{n-1}} \tag{26}$$

then, the generalization of the copy tensor follows,

$$\Delta_{a_i,a_i',a_i''}^{r_{i-1},r_i} = \begin{cases} 0 & a_i' \neq a_i'' \\ G_{a_i,a_i'}^{r_{i-1},r_i} & a_i' = a_i'' \end{cases} \tag{27}$$

there are several methods to create the tensor $\Delta^{r_{i-1},r_i}_{a_i,a'_i,a''_i}$ one of them is through the multiplication of three copy tensors and $G^{r_{i-1},r_i}_{a_i,a'_i}$. This method is shown in Fig. 3(d) where the three lines with dots in the middle represent copy tensors, while the brown tensor represents $G^{r_{i-1},r_i}_{a_i,a'_i}$.

The final representation of the generalized MPO for the multispecies GPE reads as

$$
\begin{aligned}
\mathfrak{A} = \ & \Delta^{r_1}_{a_1,a'_1,a''_1} \cdot \Delta^{r_1,r_2}_{a_2,a'_2,a''_2} \cdots \Delta^{r_{m-1}}_{a_m,a'_m,a''_m} \cdot \\
& \cdot \delta_{b_1,b'_1,b''_1} \cdot \delta_{b_2,b'_2,b''_2} \cdots \delta_{b_n,b'_n,b''_n} \cdot
\end{aligned}
\tag{28}
$$

A graphical representation of the generalized MPO $\mathfrak{A}$ and how it can be used to create the multispecies interaction term is shown in Fig. 3(b). Along with the creation procedure of $G^{r_{i-1},r_i}_{a_i,a'_i}$ in Fig. 3(d).

The last term that we need to represent is the tunneling interaction. In this case, the spatial bits are unaffected, because the interaction is linear, therefore the spatial part of the MPO will be written as several Kroenecker deltas. For the bits representing different species we have a similar, yet simpler, situation than for the interaction term. We have a tunneling tensor $\lambda^{a'_1,a'_2,...,a'_m}_{a_1,a_2,...,a_m}$ which can be transformed into an MPO with TCI. Then, the final representation of the tunneling interaction MPO can be written as

$$
\Lambda = \lambda^{r_1}_{a_1,a'_1} \lambda^{r_1,r_2}_{a_2,a'_2} \cdots \lambda^{r_{m-1}}_{a_m,a'_m} \delta_{b_1,b'_1} \delta_{b_2,b'_2} \cdots \delta_{b_n,b'_n}.
\tag{29}
$$

Figure 3(c) shows how to think about the MPO that creates the tunneling term.

In this section, we have shown how, using generalizations of the copy tensor, we can encode the evolution of a wavefunction of multiple species of BEC into a single QTT.

## 4.3  Long-Range Potentials

Another element that is interesting to introduce to the GPE is long-range potentials. They allow us to study interesting experimental effects such as breathing oscillations.

To represent long-range potentials, one often wants to model the potential created by some density distribution in $x'$ which is felt by a particle in $x$. In that case, we want to solve the following integral

$$
V_{LR}(x) = \int dx' |\psi(x')|^2 W(x,x')
\tag{30}
$$

often $W(x,x')$ represents an interaction between two particles. If interactions are translationally invariant, i.e. $W(x,x') = W(|x-x'|)$, then the QTT representation of $W(x,x')$ can be constructed in two steps. The first step is to load the single-variable function $W(y)$ into an MPS through the methods already explained in section 3.1. The second step is to contract it with the convolution operator $\mathcal{C}(y;x,x')$, such that,

$$
\begin{aligned}
W(x-x') &= \int dy\, W(y)\mathcal{C}(y;x,x') \\
&= \int dy\, W(y)\delta\left(y-(x-x')\right)
\end{aligned}
\tag{31}
$$

The convolution operator follows $\mathcal{C}(y;x,x') = \delta(y-(x-x'))$, it can be casted to a generalized MPO of bond dimension 2 with the ripple carry adder algorithm. The translation of the ripple carry adder algorithms to a tensor network is explained in detail in Appendix A. The convolution operator is a generalization of an MPO because it has three different external legs. However, the computation cost to contract it with an MPS remains small because the convolution operator has a bond dimension of 2.

445  The second term of Eq. (30) is the density profile, $|\psi(x)|^2$, which can be created following
446  the recipe explained in section 4.1. The last part to complete Eq. (30) is to perform the integral
447  of $W(x-x')|\psi(x')|^2$. As indicated in section 3, the integral of two functions that are encoded as
448  two MPSs can be computed through MPS-MPS contraction. This situation can be generalized
449  to compute the integral of a multivariable function $W(x, x')$ with $|\psi(x')|^2$ which corresponds
450  to an MPO-MPS multiplication.

# 451  5  Algorithms

452  In the following section, we present two algorithms implemented in QTT: the imaginary-time
453  evolution for finding the ground state of the GP equation, called *static solver*, and Runge-
454  Kutta-45 algorithm for the time-evolution of the GP equation with a given initial state, called
455  *dynamical solver*. Theory of these methods can be found in many references, e.g. [62], while
456  implementations are available in open-source libraries [63–71].
457  In this section, we will explain the methods we have used to find ground states and dynamic
458  solutions with the QTT formalism.

## 459  5.1  Static Solver

460  We find the ground state of the GP equation by evolving an initial trial of the ground state
461  wave function under the GP equation with imaginary time,

$$i\partial_t \psi = \hat{\mathcal{H}}_{GP}\psi \xrightarrow{\tau \to it} -\partial_\tau \psi = \hat{\mathcal{H}}_{GP}\psi \tag{32}$$

462  where $\hat{\mathcal{H}}_{GP} = \hat{H}_0 + gN|\psi(x)|^2$ where $\hat{H}_0$ is the usual quantum mechanical Hamiltonian $\hat{H}_0 = -\frac{1}{2}\nabla^2 + V(x)$.
463  As long as the initial state $\psi_0(x)$ has a non-zero overlap with the ground state, $\langle\psi_0|\psi_{gs}\rangle \neq 0$,

$$\psi_{gs}(x) = \lim_{\tau \to \infty} \frac{e^{-\tau\hat{\mathcal{H}}_{GP}}|\psi_0\rangle}{\sqrt{\langle\psi_0|e^{-2\tau\hat{\mathcal{H}}_{GP}}|\psi_0\rangle}} \tag{33}$$

464  The static solver, in this study, is derived from breaking $\tau$ in small time steps $d\tau$, and
465  creating a recursive relation by expanding Eq. (33) to first order in $d\tau$,

$$\tilde{\psi}_n(x) = \psi_{n-1}(x) - d\tau\hat{\mathcal{H}}_{GP}(x)\psi_{n-1}(x) \tag{34}$$

466

$$\psi_n(x) = \frac{\tilde{\psi}_n(x)}{\left(\int dx\,\tilde{\psi}_n^*(x)\tilde{\psi}_n(x)\right)^{\frac{1}{2}}} \tag{35}$$

467  Eq. (34) performs the imaginary time evolution, but it unnormalizes the state. Therefore, we
468  need to renormalize the state as shown in Eq. (35). With the tools explained in section 3, it
469  is straighforward to express $\mathcal{H}_{GP}$ as a generalized matrix product operator that needs to be
470  contracted with the matrix product state $\psi_{n-1}(x)$.
471  With the wavefunction obtained, we can compute its energy after $n$ time steps,

$$\begin{aligned} E &= \int dx\left(|\nabla\psi_n|^2 + V(x)|\psi_n|^2 + \frac{1}{2}gN|\psi_n|^4\right) \\ &= \int dx\left(\psi_n^*\hat{\mathcal{H}}_{GP}\psi_n - \frac{1}{2}gN|\psi_n|^4\right) \end{aligned} \tag{36}$$

472  The process is iterated until the energy converges below some tolerance, $|E_n - E_{n-1}| < \epsilon$.
473  Once the method has converged, we have found the ground state energy and the ground state
474  wavefunction for the Gross-Pitaesvkii equation.

## 5.2  Dynamic Solver

Dynamic solvers are methods that solve the time evolution of a state given initial and boundary conditions. There are a myriad of dynamic solvers that one may use to solve partial differential equations such as Euler method, Crank Nicholson, Runge-Kutta, or Arnoldi [62]. Each of those methods have an associated error which scales polynomialy with the time step $\mathcal{O}(dt^n)$. Some methods have been already implemented with Quantic Tensor Trains to solve partial differential equations [29, 40, 72–74]. In this paper, we implement the Runge-Kutta 4 (RK4) method with finite differences to the GP equation. The error committed with RK4 scales as $\mathcal{O}(dt^4)$.

RK4 is a method to solve the initial value problem,

$$\partial_t \psi = f(\psi, t); \quad \psi(t = 0) = \psi_0 \tag{37}$$

in the case of the GP equation and assuming that $\hbar = 1$, we have that

$$f(\psi, t) = -i\left(-\frac{1}{2}\partial_x^2 \psi + V(\psi, x)\psi + g|\psi|^2\psi\right) \tag{38}$$

where $V(\psi, x)$ accounts the local potential and long-range interactions. The Runge-Kutta 4 solver finds $\psi(t + dt)$ as function of $\psi(t)$ by computing

$$\psi(t + dt) = \psi(t) + \frac{dt}{6}\left(k_1 + 2k_2 + 2k_3 + k_4\right) \tag{39}$$

where

$$
\begin{aligned}
k_1 &= f\left(\psi(t), t\right) \\
k_2 &= f\left(\psi(t) + dt \cdot \frac{k_1}{2}, t + \frac{dt}{2}\right) \\
k_3 &= f\left(\psi(t) + dt \cdot \frac{k_2}{2}, t + \frac{dt}{2}\right) \\
k_4 &= f\left(\psi(t) + dt \cdot k_3, t + dt\right)
\end{aligned}
\tag{40}
$$

this means we need to calculate four different $k_i$ with QTT and then sum all of them to obtain $\psi(t + dt)$, after adding all the terms the resulting MPS $\psi(t + dt)$ may be allocating memory suboptimally. We can compress it again with a successive single value decomposition that achieves distance minimization between the original MPS, with bond dimension $\chi$, and another one with bond dimension $\chi' < \chi$, see details in [43, 75]. The steps required to create each $k_i$ are explained in the previous section (c.f. sect. 3,4). Equations (37) and (38) assume we are dealing with a single BEC species, however, the generalization to multiple species is straightforward following Eq. (7) and implementing the method explained in Sec. 4.2.

For certain simulations, we may require additional accuracy. In those cases, we could implement higher order $n$ Runge-Kutta that scale as $\mathcal{O}(dt^n)$. It is possible to use a more robust finite difference method such as HDAF, [29] which filters small machine precision errors that may pile up in high precision simulations.

## 6  Applications

In the following, we apply the QTT methods that have been explained in previous sections to solve applications for the one-dimensional GP equation. In particular, we study the mean-field ground state of the BEC in a harmonic trap, dynamics of a BEC in the presence of quasi-disorder potential, spinor BEC dynamics, as well as BEC dynamics with long-range interactions. We denote the free part of the GP equation as $\hat{H}_0 = -\frac{1}{2}\partial_x^2 + \frac{1}{2}x^2$.

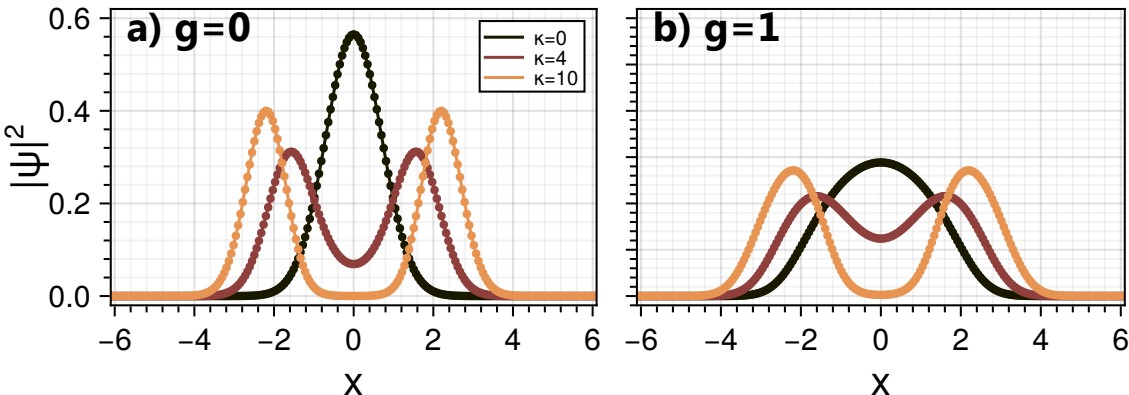

Figure 4: Ground state of the GP equation with different interaction strengths ($g$) and different barrier heights of the double well potential ($\kappa$). From left to right we present the ground state wave-function of no interaction $g = 0$, and a repulsive interaction $g > 0$ with different barrier heights ($\kappa = 0, 4, 10$).

## 6.1  Ground state

We start by considering bosons in a harmonic trap experiencing a Gaussian barrier with an amplitude $\kappa$ located at the trap center $x = 0$. The free Hamiltonian reads $\hat{H}_0 = -\frac{1}{2}\partial_x^2 + \frac{1}{2}x^2 + \kappa \exp\left(-\frac{x^2}{2}\right)$. For large $\kappa \gg 1$ bosons are trapped in a double well potential.

We calculate the ground state of the interacting gas in repulsive regime $g = 1$ and non-interacting regime $g = 0$. With the help of imaginary time evolution, described in Sec. 5.1, we find the lowest energy eigenstate, starting from a Thomas-Fermi profile

$$|\psi_{TF}(x)|^2 = \begin{cases} \mathcal{N}\frac{R_{TF}^2-x^2}{2N} & |x| < R_{TF} \\ 0 & |x| > R_{TF} \end{cases} \tag{41}$$

with $R_{TF}^2 = 2\mu$, where $\mu$ is the chemical potential given by the parameters of the system, and $\mathcal{N}$ is the normalization constant.

Fig. 4 presents ground state obtained for $g = 0$ Fig. 4(a) and $g = 1$ Fig. 4(b) for central barrier amplitude $\kappa = 0, 4, 10$. For $\kappa = 0$ the non-interacting ground state is a simple Gaussian wave-packet, i.e. lowest eigenstate of the Harmonic Oscillator, while for repulsive interactions, $g = 1$, the ground state is close to the Thomas-Fermi profile, with a final healing length. For $\kappa = 4$ atoms in ground state localize in the left/right well, with non-zero probability in the trap center at $x = 0$. For a strong barrier, $\kappa = 10$, bosons are localized in left/right well and exponentially vanishing probability density in the trap center $x = 0$. The QTT results are in full agreement up to numerical precision with standard GP solvers.

## 6.2  Quasi-disorder potential: Aubry-André-Harper model

In the following, we employ QTT method to study the interplay between interactions and disorder potential in BEC. In the non-interacting scenario, when BEC experiences only disorder potential, in quasi-one dimensional geometry Anderson localization of matter waves occurs, experimentally observed for speckle disorder potentials [76].

The Aubry-André-Harper (AAH) model [77, 78] is defined as a cosine function with an irrational spatial modulation breaking its periodicity. AAH model is a quasi-random potential, with an infinite range correlation length, considered as a toy model allowing studies on localization-delocalization transition with a pseudo-disorder potential, giving rise to such phenomena as multifractality [79]. Localization-delocalization transition in AAH model in BEC has been studied experimentally with quasi-disorder potential [80–84].

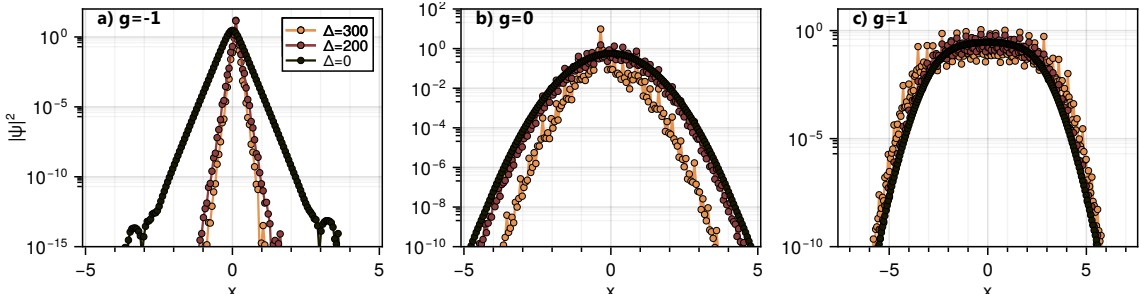

Figure 5: Ground state of the GP equation with different interaction strength ($g$) and different disorder strength ($\Delta$). From left to right we present the ground state wave-function of an attractive interaction $g < 0$, no interaction $g = 0$ and a repulsive interaction $g > 0$ with different disorder strengths.

In the following, we use QTT to numerically study the interplay between mean-field interactions and quasi-disorder potential on the dynamics of BEC, governed by:

$$i\partial_t \psi(x,t) = \left(\hat{H}_0 + V_{AAH} + g_{1D}N|\psi(x,t)|^2\right)\psi(x,t), \tag{42}$$

where quasi-disorder is given by

$$V_{AAH} = \Delta \cos\left(2\pi\beta(x+L)\right) \tag{43}$$

where $\beta = \frac{\sqrt{5}-1}{2}$ is irrational number resulting in quasi-periodicity, $\Delta$ is quasi-disorder amplitude, and $\delta x$ is the numerical space discretization, $x \in [-L, L]$.

We prepare the ground state of the BEC, Eq. (42), with imaginary time evolution as it is explained in section 5.1. Next, we switch off the harmonic potential and evolve the system in the presence of both interactions and quasi-disorder potential.

The results of the ground state densities wave-functions are shown in Fig. 5, for $g = -1, 0, 1$ and $\Delta = 0, 200, 300$. Fig.5(a) presents exponentially localized profiles for attractive interactions, $g = -1$. The presence of the quasi-disorder potential, $\Delta = 200, 300$, supports the exponential localization of the wavepacket, observed as a smaller localization length compared to the case without quasi-disordered potential, $\Delta = 0$. For non-attractive interactions $g = 0, 1$, panels (b), and (c) respectively. In both scenarios we see that AAH weakly affects the ground state density profile, mainly determined by the presence of the harmonic trap.

Next, we study the time evolution after switching off the harmonic trap. Fig. 6, panels (a)-(i) present evolution of the density profile for given $(g, \Delta)$. A quantitative comparison of the BEC dynamics can be provided by the density width evolution $\sigma^2(t) = \langle x^2 \rangle - \langle x \rangle^2$, $\langle A \rangle = \int A|\psi(x,t)|^2 dx$, presented at bottom panel on Fig.6 For attractive interactions, $g = -1$ (panels (a), (b), (c)), the system remains localized in the center of the system, and $\sigma(t)$ is constant. For the non-interacting scenario, $g = 0$ (panels (d), (e), (f)), evolution has ballistic behavior: for free-evolution $\Delta = 0$ Fig. 6(d) the width of the BEC scales linearly with time as $\sigma \propto t$. The presence of the quasi-disorder, ($\Delta = 200, 300$), prevents spread of the wavefunction, localizing the system, which results in $\sigma(t) = const.$. The most interesting scenario corresponds to interplay between repulsive interactions, and the quasi-disorder potential. For $\Delta = 0, 200$ system spreads, indicated by monotonically increasing $\sigma(t)$, however for strong enough $\Delta = 300$, the wave-packet evolution is dominated by the quasi-disorder potential and after initial spread, wave function dynamics is trapped.

## 6.3 Long-range interactions: Rydberg-dressing

In [85], authors proposed an experimental protocol to observe the effect of induced long-range Rydberg-dressed interactions on a Bose-Einstein condensate. The Rydberg-dressing lasers in-

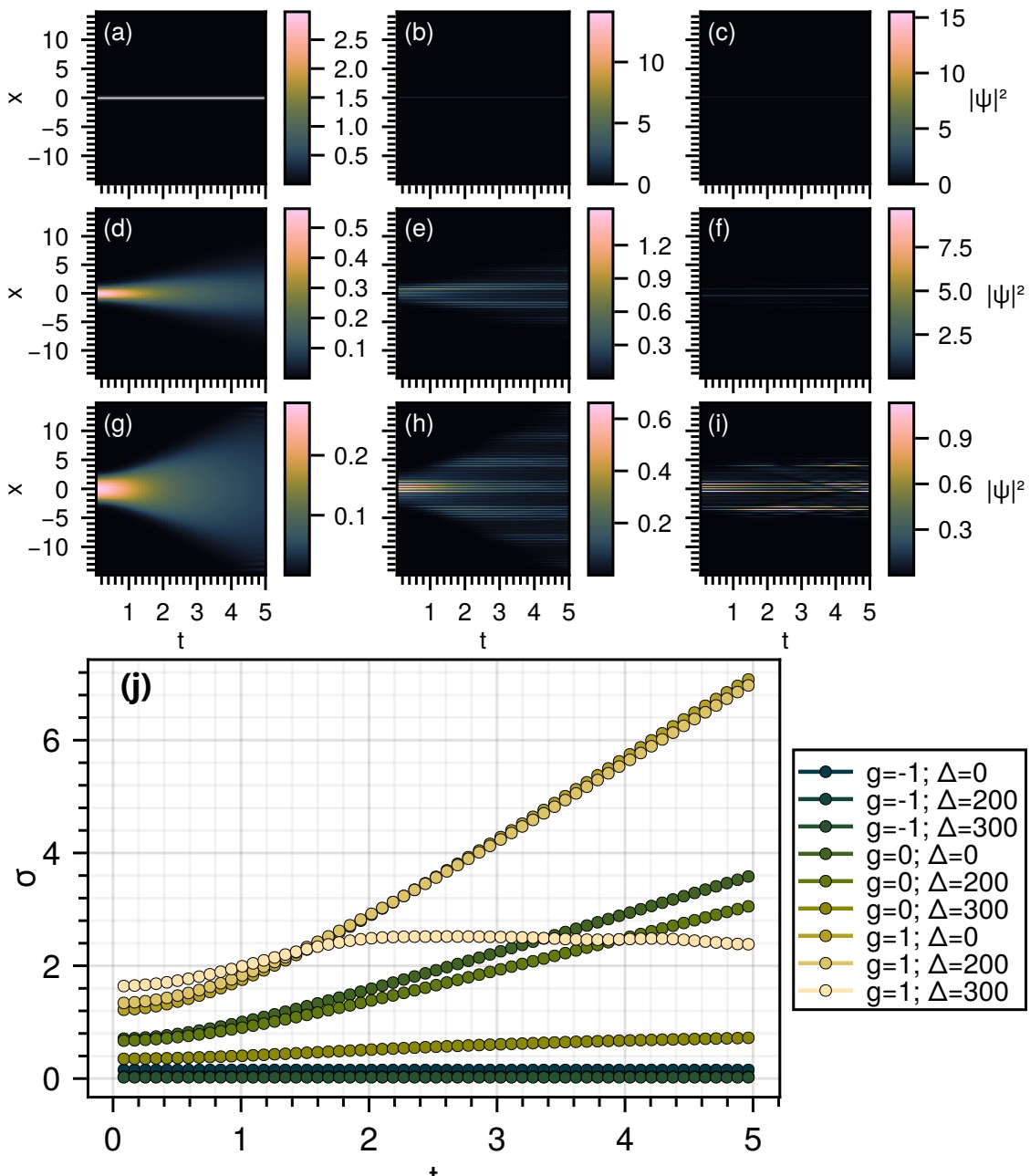

Figure 6: Panels (a)-(i): color encoded density of the wave-packet evolution $|\psi(x,t)|^2$, initially prepared in a ground state with a harmonic trap, for a different set of parameters $\{g, \Delta\}$. Different rows represent different interaction strengths $g = -1, 0, 1$ respectively, while different columns correspond to different disorder strengths $\Delta = 0, 200, 300$. The bottom panel (j) presents the evolution of the spatial standard deviation with time for the GP equation with disorder and different interaction strengths. The dark green lines represent a BEC with repulsive interactions ($g < 0$), the light green lines represent a non-interacting BEC ($g = 0$), and the yellow lines represent repulsive interactions ($g > 0$). For each case, we have studied 3 different disorder strengths $\Delta = 0, 200, 300$.

duce long-range interactions between two ground-state atoms

$$W(r) = \beta^4 \frac{C_6}{R_B^6 + r^6}, \tag{44}$$

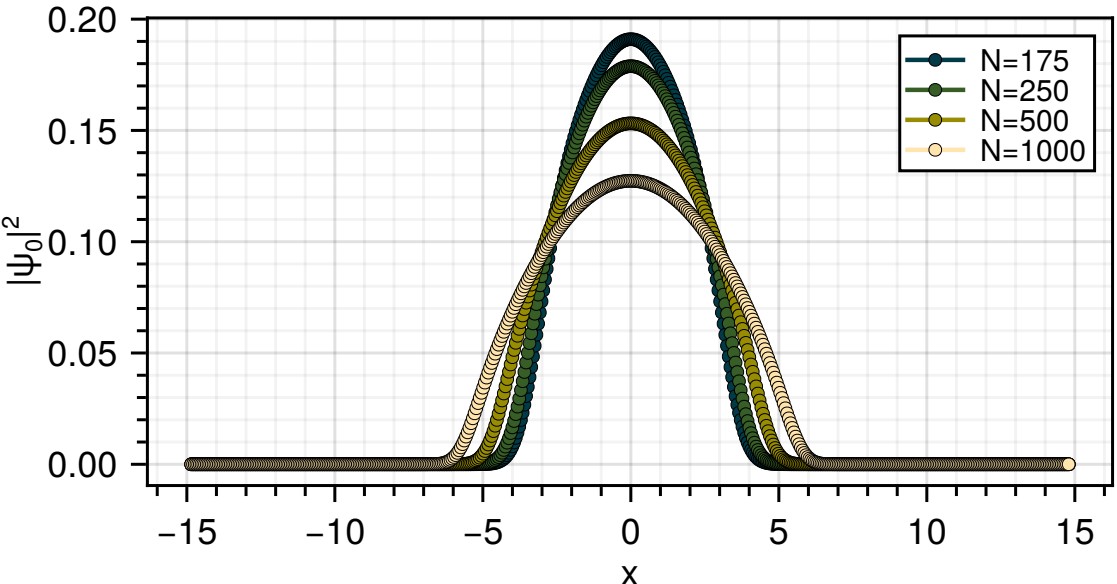

Figure 7: Ground state probability density of the GP equation with Rydberg-dressed interactions ($\beta = 2$) for $N = 175, 250, 500, 1000$ particles.

where $\beta \ll 1$ is an effective Ryderg-dressing laser detuning, $C_6$ is a van-der Waals interaction coefficient, and $R_B^6$ is an effective Rydberg-blockade radius; parameters are determined by the experimental conditions [86]. In the regime where the radius of the BEC is smaller than the blockade radius, the Rydberg-dressed interactions result in an effective non-linear potential

$$V_{LR}(x) = \int |\psi(x')|^2 W(x - x') \mathrm{d}x'. \tag{45}$$

The protocol starts with preparing the BEC in the ground state in a harmonic trap with the Rydberg-dressing lasers switched-on. Next, the Rydberg-dressing lasers are switched-off, and the system evolves under GP equation, with short-range interactions. After a quench, an effective trapping frequency experienced by BEC is rapidly changed, and the BEC performs a breathing oscillation, observed as time-periodic change of the BEC wave-packed width, where oscillation frequency depends on $\beta$, while its amplitude on number of particles $N$.

We study the abovementioned protocol with our numerical method. We start with an imaginary-time evolution to find the ground state, considering a fixed value of Rydberg-dressing parameter $\beta$, and different particle numbers $N = 175, 200, 500, 1000$. Changing the number of particles in the BEC effectively changes the mean-field strength of the interactions between the particles in it. Therefore, as we see in Fig. 7, for larger particle numbers the ground state wavefunction is more spread along our domain.

After the ground state has been found, then we perform a real-time evolution under the GP equation without long-range interactions, which corresponds to turning-off the Rydberg-dressing lasers, inducing a breathing mode to the BEC. The breathing modes can be seen in a two-dimensional plot of the density of the bosons with time Fig. 8(a-d). However, the breathing mode is better characterized by computing the normalized spread of the wavefunction with time ($\sigma(t)/\sigma(0)$) so we can compare the results for different particle numbers. The change of the BEC width oscillates in time, with a maximal value at $t = 0$, see Fig. 8. The oscillations period depends on dressing parameter $\beta$, while its amplitude depend on particle number $N$ – these results agree with other studies that have found the similar behavior regarding the breathing oscillations, [85].

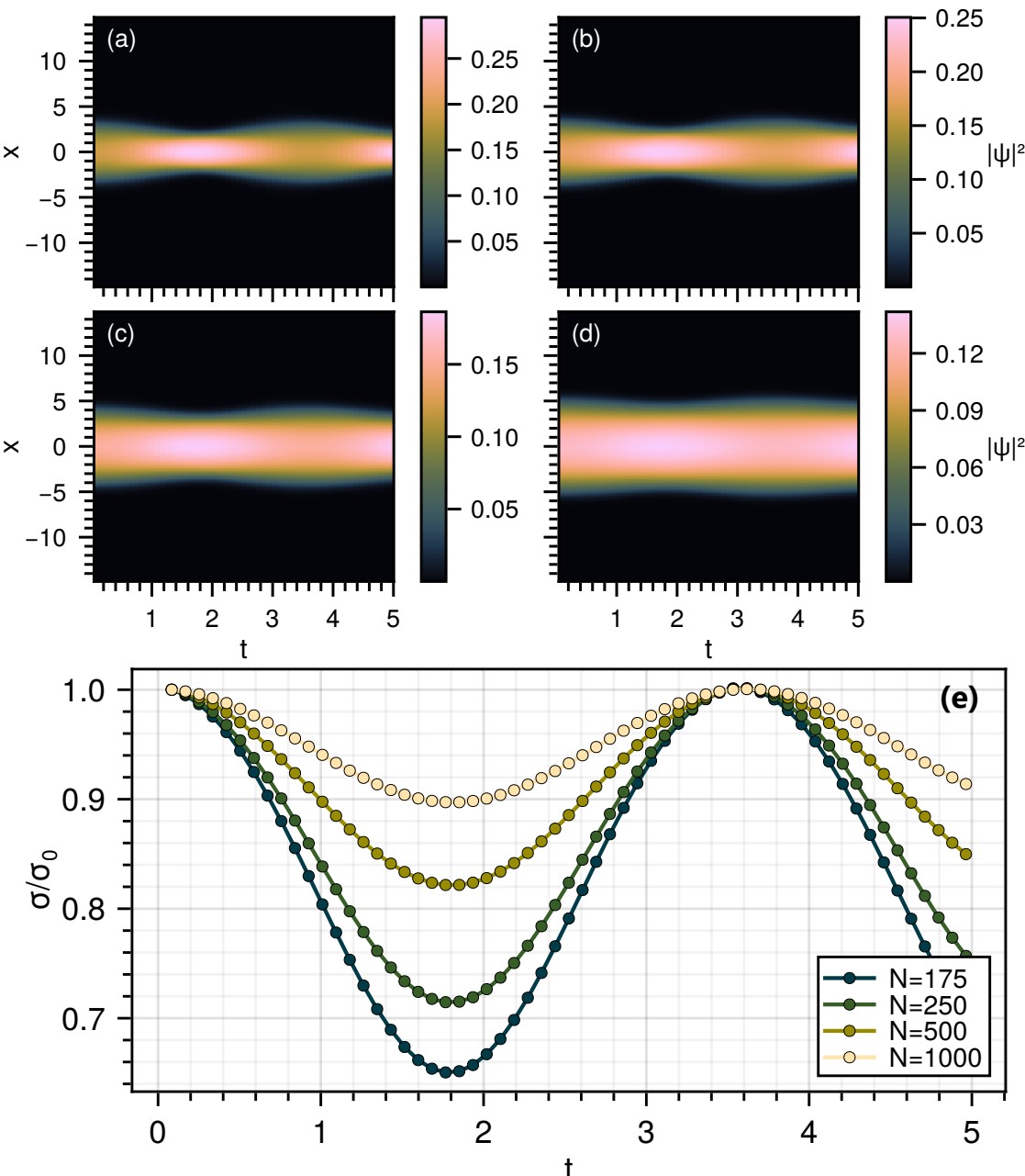

Figure 8: Panels (a)-(d) present color-encoded evolution of the probability density $|\psi(x,t)|^2$ after switching-off Rydberg-dressing lasers. BEC dynamics is defined by the breathing mode, with frequency depending on dressing parameter $\beta$, with amplitude depending of particle number $N$. Here $N = 175, 250, 500, 1000$ for panels (a)-(d), respectively. Bottom panel presets evolution of the relative BEC wave-packet width $\sigma(t)/\sigma(t=0)$, where $\sigma(t=0)$ corresponds to BEC ground state width in the presence of Rydber-dressed interactions.

## 6.4 Two-component BEC

In the following, we study the dynamics of $K$-component one-dimensional BECs described by the set of coupled GP equations [87], Eq.(7). Where $g_{kj}$ describes the inter- and intra-component interactions, while $\lambda$ describes the amplitude of *tunneling* a particle between components. For this experiment we assume $g_{ij} = g_{ji}$, therefore there will be no phase difference between species.

Fig. 9 shows the time evolution assuming only two species of interacting BEC $K = 2$. For all the simulations, $g_{11} = g_{22} = 1$ and $g_{12} = g$. We consider the initial wave-function as

$$\psi_k(x,0) = \frac{1}{2}\text{sech}\left(x + (-1)^k \cdot 10\right) e^{i(-1)^k \cdot \frac{p_0}{4} x}, \tag{46}$$

which corresponds to two localized BEC, each one centered at $x = \pm 10$ with initial momentum $\pm\frac{p_0}{4}$. We consider three different non-fixed parameters: $\lambda, g$, and $k_0$; which we will vary to assess the importance of each of them individually.

Figs. 9(a,b) correspond to the evolution of the two species inside a trapping potential and repulsive interaction, without tunneling between spieces ($\lambda = 0$), no inter-species interaction ($g = 0$), and zero initial momentum ($k_0 = 0$). The behavior of both species is symmetric, they oscillate around the symmetry point of the trapping potential $x = 0$.

Figs. 9(c,d) correspond to non-zero initial mommentum, $k_0 = 40$. The wave-function of both species is still symmetric, but now it travels farther away from the center of the potential. However, as expected, the period of oscillation around the trapping potential remains the same and it is independent of the initial momentum.

Figs. 9(e,f) show the behavior when tunneling between species is allowed, $\lambda = 0.5$. In this case, the oscillations are still visible, but now the species pollute the original trajectory with some support in the trajectory of the other species. Additionally, when the BEC arrives near the point $x = 0$ an interference pattern is formed because the are two branches of the wavefunction meeting at the same point. Although in this case, the pattern is a bit different with respect to the other ones, the wavefunctions still display the symmetry between species and the same period as in the other cases.

The last case of study is when the interaction term is non-zero ($g = 60$). In this case, we expect repulsion when both species are near each other. However, in this experiment, both species meet briefly and it is difficult to observe an appreciable difference in the wavefunction of both species, Figs. 9(g,h).

The presented results obtained with the proposed QTT method, are in full agreement with results presented in [87], obtained with standard methods for GP equations.

## 7 Numerical resources

Standard numerical resources for solving GP equation in one-dimensional geometry require $N_{\text{grid}}$ complex parameters to store a wave-function on a domain $L$ with $N_{\text{grid}}$ discretization points, while each matrix-vector multiplication requires $N_{\text{grid}}^2$ operations in a dense representation. Choosing discretization size $N_{\text{grid}}$ in a standard approach depends on physical parameters of the system, such as number domain size $L$, number of particles $N$, and interaction strength amplitude $|g|$. In particular, faithful representation of an interacting BEC in a harmonic trap requires the spatial discretization step $\delta x = L/N$ to be much smaller than characteristic length scale of the system given by the healing length $\xi = 1/\sqrt{2\mu}$ ($\mu = |g|N/L$), i.e. $\xi/\delta x \gg 1$. Next, the spatial discretization step determines the time-discretization step required for stable time evolution, i.e. $dt \ll \delta x^2$ [62].

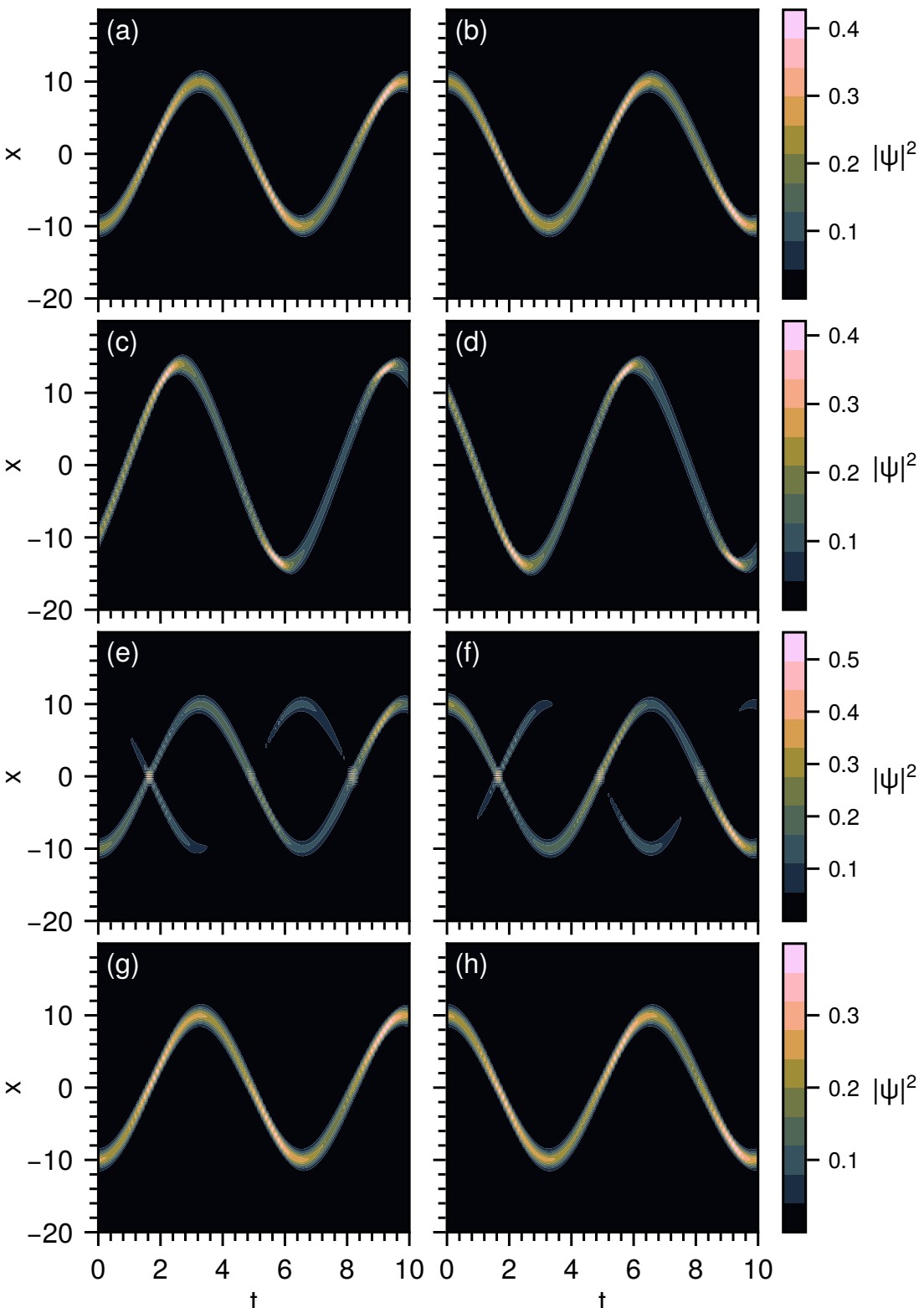

Figure 9: Time evolution of the absolute value of the wave-function of a each species,$|\psi_a(x,t)|$ (right and left), evolving under the coupled GP equations in Eq. (7) different combinations of the free parameters $g$, $k_0$, and $\lambda$. Panels (a)-(b) correspond to $\lambda = p_0 = g = 0$. Panels (c)-(d) correspond to $\lambda = g = 0$, $p_0 = 40$. Panels (e)-(f) correspond to $p_0 = g = 0$, $\lambda = 0.5$. Panels (g)-(h) correspond to $p_0 = \lambda = 0$, $g = 60$.

In section 3 we have briefly discussed that QTT may be able to reduce both the computational and the storage cost of the solution of the GP equation. In general, the number of parameters required to represent an MPS is upper bounded by $\mathcal{O}\left(2\log_2(N_{\text{grid}})\chi^2\right)$, and the number of operations to calculate an MPS-MPO contraction is upper bounded by $\mathcal{O}\left(4\log_2(N_{\text{grid}})(\chi^3 k + \chi^2 k^2)\right)$, where $\chi$ is the maximum bond dimension of the MPS and $k$ is the maximum bond dimension of the MPO. Such computational and storage cost needs to be compared with traditional methods.

In the following, we discuss the concrete example of the ground state calculations from section 6.1 to display how QTT improves the computational capacity of traditional tensor networks without deteriorating the final results. In the case of $\kappa = 10$ and $g = 1$, we can faithfully represent the ground state wavefunction with $N_{\text{grid}} = 512 = 2^9$ grid points. The same ground state wavefunction with QTT has maximum bond dimension $\chi = 8$, thus the number of parameters required to store the wavefunctions as an MPS is $\mathcal{O}\left(2\log_2(N_{\text{grid}})\chi^2\right) = \mathcal{O}(1152)$. In this case, storing the wavefunction as an MPS is not more effective than doing so with traditional methods, but it was to be expected since QTT formalism is expected to surpass traditional methods when a large number of points is required. For example, the same example has been run for $N_{\text{grid}} = 4096 = 2^{12}$ grid points. Traditionally, we require 4096 to store the wavefunction. However, with QTT the maximum bond dimension required is $\chi = 9$, thus the number of parameters required to represent the MPS is upper-bounded by $\mathcal{O}(1944)$. For this case, we can see that storing the ground state wavefunction as a QTT is more efficient than using traditional methods.

Fig. 10(a) shows the number of parameters to store a ground state wavefunction with traditional methods vs QTT for different grid discretization. As expected, the cost increases linearly with the number of grid points for traditional methods, while it scales sublinearly for QTT. We find that for the precise wavefunction displayed in Fig. 10(a) the QTT representation becomes efficient for $N_{\text{grid}} > 2^{10}$ grid points. The last point of the QTT line in Fig. 10(a) has a small increase that can be interpreted as if it were to start displaying a linear scaling. However, that is not the case, the reason for this increase is that for this particular number of grid points, the maximum bond dimension has increased from $\chi = 8$ to $\chi = 9$, therefore there is a larger increase in the number of parameters. However, as discussed in section 3, as long as the bond dimension increases mildly with the number of bits, the compression remains efficient, as displayed in the figure.

The computational cost in this example is a little more subtle. Since the matrix representation of a Hamiltonian $\hat{H}_0$ can be represented as a sparse matrix, the number of operations required to find $\psi_i(x)$ from $\psi_{i-1}(x)$ is reduced from $\mathcal{O}\left(N_{\text{grid}}^2\right)$ to $\mathcal{O}\left(3N_{\text{grid}}\right)$. Thus, for $N_{\text{grid}} = 4096$, the number of operations required, to take an additional step of the static solved defined in section 5.1, is $\mathcal{O}\left(10^4\right)$. On the other hand, the number of operations required in the QTT formalism is upper-bounded by $\mathcal{O}\left(4\log_2(N_{\text{grid}})\left(\chi^3 k + k^2\chi^2\right)\right) \approx \mathcal{O}\left(10^5\right)$, because the maximum bond dimension of $\hat{H}_0$ is $k = 8$. Due to the fact that traditional methods are able to use sparse matrices to reduce the number of operations required to perform matrix-vector multiplication, they are more optimal than QTT.

Fig. 10(b) shows the number of operations required to perform a matrix-vector or MPO-MPS multiplication. We can see that traditional methods scale polynomially with the number of grid points of the system, while the QTT method scales sub-polynomially (possibly logarithmically) with the grid points of the system, and while the sparse traditional methods require a smaller number of operations we can imagine that for a large enough number of grid points, the QTT method will be more efficient than the sparse method. Curiously, the increase in the last point of the QTT line in 10(b) is milder than in (a) because although the bond dimension of the wavefunction increases from $\chi = 8$ to $\chi = 9$, the bond dimension of the MPO decreases from $k = 9$ to $k = 8$, that is why the number of operations remains roughly the same.

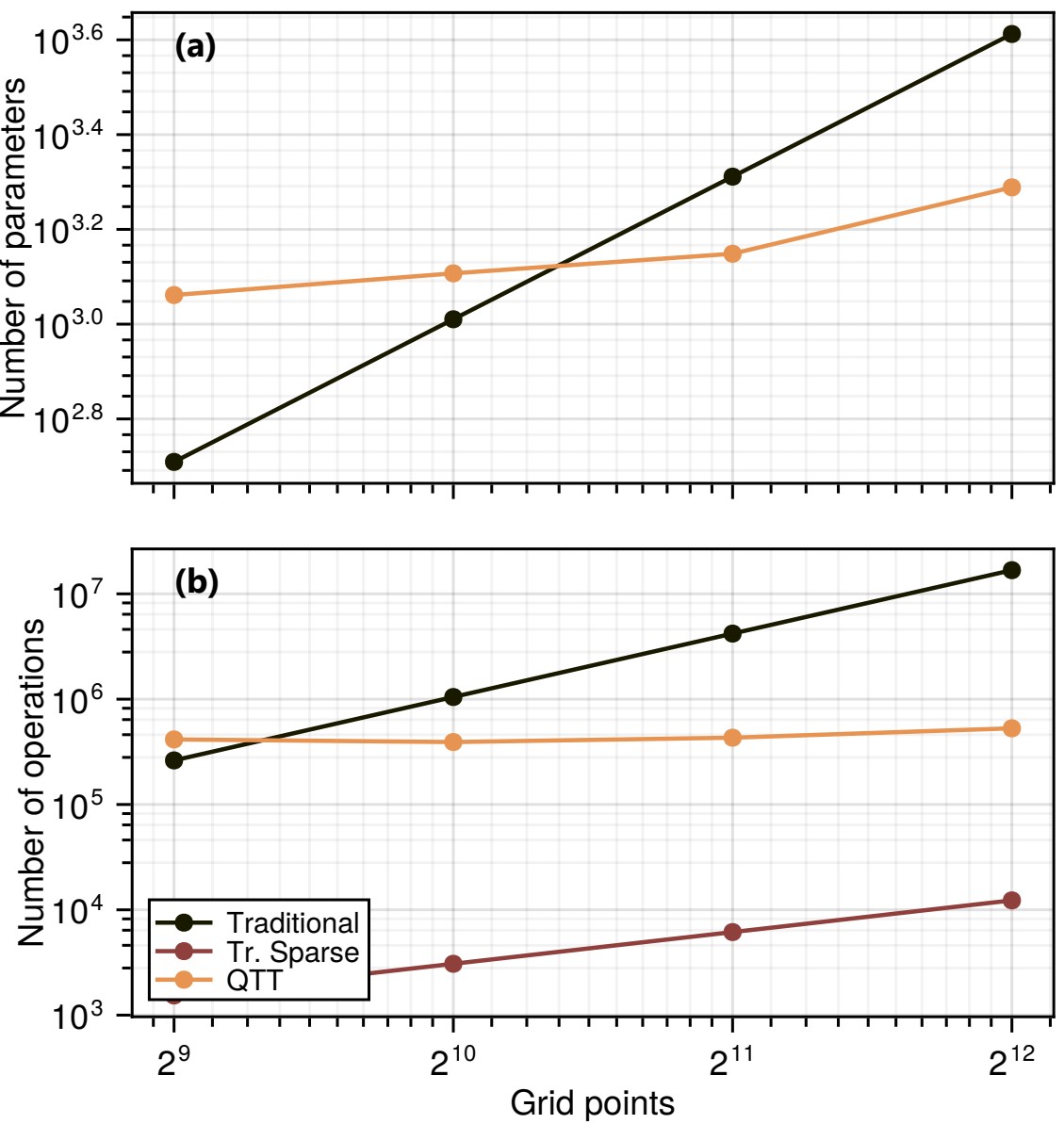

Figure 10: Storage (a) and computational (b) cost of storing the ground state wave-function shown in Fig. 4(b) for $\kappa = 10$ and different number of grid points. Figure (a) displays the number of parameters required to store the wavefunction of the GP equation with a double well potential by traditional methods (black) and QTT (orange). Figure (b) shows the number of operations required to perform the operation $\hat{H}_0|\psi\rangle$ (Eq. (34)) which corresponds to a matrix-vector multiplication or an MPO-MPS contraction. For a matrix-vector multiplication using dense algorithms (black), the matrix-vector multiplication with a sparse matrix (red), and MPO-MPS multiplication with QTT (orange).

The computational gain of using QTT for GP equation, in terms of memory resources, is more visible in higher dimensional geometries. In particular, in three-dimensions, discretization of domain $L^3$ requires $N_{\text{grid}}^3$ grid points, while in QTT it scales linearly as $\propto \mathcal{O}(3N_{\text{grid}})$. Computational gain is even more visible when considering interacting systems of $M$-indepenent BECs in $d$ dimensional geometry each. The standard approach requires $N_{\text{grid}}^{Md}$ grid points to store $M$ $d$-dimensional wavefunctions, while QTT resources scale linearly as $\mathcal{O}(MdN_{\text{grid}})$.

# 8 Conclusions

In summary, we have introduced a tensor train formulation of the one-dimensional Gross–Pitaevskii equation that compresses both the condensate wave function and the nonlinear operators acting on it into logarithmically scaled tensor cores. Within this representation we proposed an imaginary-time projector that converges to the variational ground state and a fourth-order Runge–Kutta integrator that advances the system in real-time while actively truncating QTT ranks under a controlled error threshold. Because every step remains inside the low-rank manifold, the computational cost grows only polylogarithmically with the number of spatial grid points. We benchmark our results with standard methods for GP equation, obtaining agreement on the numerical precision in each considered case.

Our results further develop the tensor network toolbox to treat non-linearities, long-range interactions, and multispecies BECs by introducing small modifications of the copy tensor. In this work, tensor networks are presented as a practical, quantum-inspired alternative to conventional methods to solve partial differential equations and to nascent quantum-computing approaches for the numerical solution of partial differential equations. The tensor networks are dimension-agnostic and naturally parallelizable, suggesting straightforward extensions to two- and three-dimensional GPEs, multicomponent mixtures, and finite-temperature stochastic variants. Beyond cold-atom physics, the same ideas can be transplanted to nonlinear optical envelopes, plasma wave packets, and other nonlinear Schrödinger-type problems. We, therefore, anticipate QTT techniques becoming a valuable addition to the standard numerical toolbox wherever long propagation distances, fine spatial structure, or large parameter sweeps overwhelm traditional mesh-based solvers.

**Funding information**   ABC is supported by Grant MMT24-IFF-01. The funding for this action/grant and contracts comes from the European Union's Recovery and Resilience Facility-Next Generation, in the framework of the General Invitation of the Spanish Government's public business entity Red.es to participate in talent attraction and retention programs within Investment 4 of Component 19 of the Recovery, Transformation, and Resilience Plan. MP acknowledges support from: European Research Council AdG NOQIA; MCIN/AEI (PGC2018-0910.13039/501100011033, CEX2019-000910-S/10.13039/501100011033, Plan National FIDEUA PID2019-106901GB-I00, Plan National STAMEENA PID2022-139099NB, I00, project funded by MCIN/AEI/10.13039/501100011033 and by the "European Union NextGenerationEU/PRTR" (PRTR-C17.I1), FPI); QUANTERA DYNAMITE PCI2022-132919, QuantERA II Programme co-funded by European Union's Horizon 2020 program under Grant Agreement No 101017733; Ministry for Digital Transformation and of Civil Service of the Spanish Government through the QUANTUM ENIA project call - Quantum Spain project, and by the European Union through the Recovery, Transformation and Resilience Plan - NextGenerationEU within the framework of the Digital Spain 2026 Agenda; Fundació Cellex; Fundació Mir-Puig; Generalitat de Catalunya (European Social Fund FEDER and CERCA program; Barcelona Supercomputing Center MareNostrum (FI-2023-3-0024); Funded by the European Union. Views and opinions expressed are however those of the author(s) only and do not necessarily reflect those of the European Union,

European Commission, European Climate, Infrastructure and Environment Executive Agency (CINEA), or any other granting authority. Neither the European Union nor any granting authority can be held responsible for them (HORIZON-CL4-2022-QUANTUM-02-SGA PASQuanS2.1, 101113690, EU Horizon 2020 FET-OPEN OPTOlogic, Grant No 899794, QU-ATTO, 101168628), EU Horizon Europe Program (This project has received funding from the European Union's Horizon Europe research and innovation program under grant agreement No 101080086 NeQSTGrant Agreement 101080086 — NeQST); ICFO Internal "QuantumGaudi" project.

# A  Translation and Convolution Operators

The translation operator can be cast into a matrix product state with the QTT formalism by using the *ripple carry adder* algorithm. The algorithm just performs the elementary bit addition operator and in case the sum is larger than 1, it carries the rest into the next bit. The first operator we want to encode is the translation operator, given a function and a real number $a \in \mathbb{R}$ we want to find a matrix product operator $(\mathcal{T}_a)$ such that $f(x) \xrightarrow{\mathcal{T}_a} f(x+a)$. Assuming that $a = \sum_{i=1}^{n} a_i 2^{-i}$, then we need to create two different tensors, $T^{a_i=0}$ and $T^{a_i=1}$. $T^{a_i}$ represent the smaller tensors that conform the MPO $\mathcal{T}_a$:

$$\mathcal{T}_a = \sum_{b_i} \sum_{b_i'} T^{a_1}_{b_1, b_1', l_1} T^{a_2}_{l_1 b_2, b_2', l_2} \dots T^{a_n}_{l_{n-1} b_n, b_n'} |\{b_i'\}\rangle\langle\{b_i\}|, \tag{A.1}$$

where $|\{b_i\}\rangle = |b_1, b_2, \dots b_n\rangle$. Each tensor $T^{a_i}_{l_{i-1}, b_i, b_i', l_i}$ has 4 important indices, $l_i$ is the index that informs you if you carry any units from the previous operation. $b_i$ is the term that gets added to $a_i$, $b_i'$ is the resulting term from the operation $b_i' = (a_i + b_i) \mod 2$ and $l_{i-1}$ will carry the residual unit in case it is needed. Recall that we assume that $b_1$ corresponds to the largest bit, $2^{-1}$, and $b_n$ the smallest, $2^{-n}$. The set of indices $\{l_i\}_{i \in \mathcal{I}_1}$ are link indices while $\{b_i\}_{i \in \mathcal{I}_2}$ correspond to the spatial indices (physical indices).

Once $l_i$ and $b_i$ are fixed, we have enough information to know all the non-zero terms of $T^{a_i}_{l_{i-1}, b_i, b_i', l_i}$. The only non-zero terms of $T^{a_i}_{l_{i-1}, b_i, b_i', l_i}$ are the following,

$$\begin{aligned} T^{(a_i=0)}_{0,0,0,0} = T^{(a_i=0)}_{0,0,1,1} = T^{(a_i=0)}_{0,1,1,0} = T^{(a_i=0)}_{1,1,0,1} = 1 \\ T^{(a_i=1)}_{0,0,1,0} = T^{(a_i=1)}_{1,0,0,1} = T^{(a_i=1)}_{1,1,0,0} = T^{(a_i=1)}_{1,1,1,1} = 1 \end{aligned} \tag{A.2}$$

The first line corresponds to the tensors in which $a_i = 0$. From left to right, the first sum is the addition of $(0 + 0 + 0) \mod 2 = 0$, the second and the third is $(1 + 0 + 0) \mod 2 = (0 + 1 + 0) \mod 2 = 1$, and the last $(1 + 1 + 0) \mod 2 = 0$ but in this case we need to carry one towards the next tensor. The second line corresponds to the tensors in which $a_i = 1$. From left to right, the first sum corresponds to $(0 + 0 + 1) \mod 2 = 1$, the second and the third $(1 + 0 + 1) \mod 2 = (0 + 1 + 1) \mod 2 = 0$ but we need to carry one over the next tensor, and the last $(1 + 1 + 1) \mod 2 = 1$ and we carry one to the next tensor.

After the creation of both tensors depending on each value $a_i$, we can construct the MPO tensor by tensor. However, there are two links that are not contracted, $l_0$ and $l_n$. Depending on the vector we contract these links with, we will have periodic or open boundary conditions of the sum. $l_n$ always needs to be contracted with the vector $[1, 0]$ because we do not start the sum carrying a 1. Then if $l_0$ is contracted with $[1, 0]$, we will impose open boundary conditions; while if we contract it with $[1, 1]$ we will impose periodic boundary conditions. From the translation operator is easy to create the inverse translation operator by applying the adjoint to $\mathcal{T}_{-a} = (\mathcal{T}_a)^\dagger$. Figure 11 shows a graphical representation of both $\mathcal{T}_a$ and $\mathcal{T}_{-a}$.

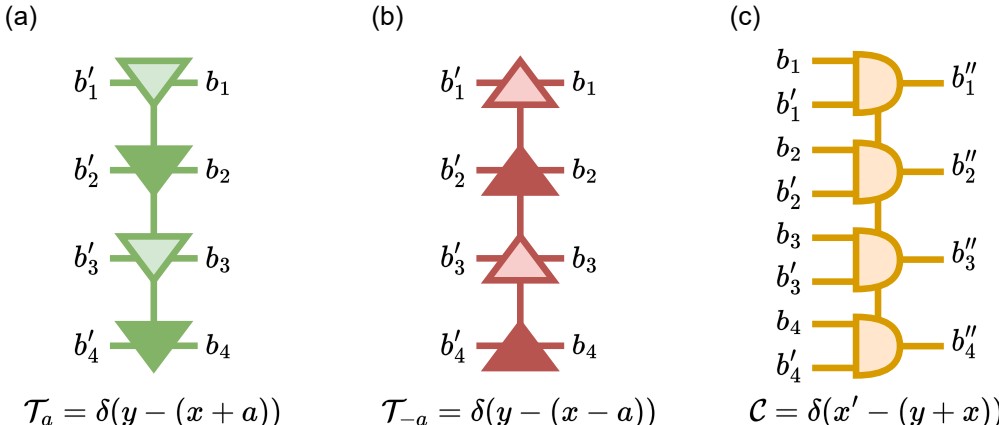

Figure 11: The panels (a) and (b) depict a positive and negative translation operator. In this case $a = \frac{1}{4} + \frac{1}{16}$, the filled (empty) triangles mean that $a_i = 1$ ($a_i = 0$). Per convention, we choose that the triangles pointing down (up) will represent the positive translation(negative) translation. Fig. 2(a) shows how we create the differential operator from two translation operators. Panel (c) shows a graphical representation of the generalized MPO required to represent the convolution operator.

The translation operators, as has been discussed in section 3, creates the matrix product operator representation of the $\delta$-function,

$$f(x \pm a) = \int \mathrm{d}y\, \delta(y - (x \pm a)) f(y) = \mathcal{T}_{\pm a} f^{QTT} \tag{A.3}$$

this can be further generalized by adding a new index that works for any $a \in [0, 1)$. For the case where $a$ is a new variable, it is more usual to call it $a = x'$. The objective now is to create a generalized matrix product operator (gMPO) that allows you to create from $W(x) \xrightarrow{\mathcal{C}} W(x - x')$. This generalized MPO will have three physical indexes instead of two and will encode $\delta(y - (x - x'))$, recall that depending on what index you are contracting the function with, one could also create $W(y + x')$. In Fig. 11(c) there is a graphic representation of the generalized MPO.

The idea to create $\delta(y - (x - x'))$ is the same as in the translation case. For self-consistency, we will create $\delta(x - (y + x'))$ but then we will contract the tensor $W^{MPS}(y)$ with the variable $y$ effectively creating $W(x - x')$. The tensors conforming the convolution operator, $\mathcal{C}$, are $C_{l_{i-1}, b_i, b'_i, b''_i, l_i}$, as before $l_i$ represent bond dimensions and $b_i, b'_i, b''_i$ represent the spatial bits.

Following the ripple carry algorithm we have that the only non-zero term of $C_{l_{i-1}, b_i, b'_i, b''_i, l_i}$ are,

$$\begin{aligned}
C_{0,0,0,0,0} = C_{0,0,1,0,1} = C_{0,1,1,0,0} = C_{1,1,0,0,1} = 1 \\
C_{0,0,1,1,0} = C_{1,0,0,1,1} = C_{1,1,0,1,0} = C_{1,1,1,1,1} = 1
\end{aligned} \tag{A.4}$$

as before there will be two dangling bonds $l_0$ and $l_n$ that can be contracted to impose periodic or open boundary conditions with the same vectors as before. The convolution operator gives us the option of imposing exact translation symmetry in our tensor and creating the functions $W(x - x')$. Such tensors are used in many areas of physics, in our case we use the convolution operator to compute the mean-field dipolar interaction of a BEC, Eq. (30).

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
