# Peer review of "Quantics Tensor Train for solving Gross-Pitaevskii equation"

_SciPost Physics_

## Round 1 · Referee Report · Anonymous (Referee 1) · 2025-7-16

Strengths

  1. The present article addresses a timely question in computational physics: using novel, quantum-inspired tools to solve partial differential equations (PDE), in particular PDEs with the notoriously hard non-linearities, such as exemplified by the GPE.

  2. It is well-written in an approachable language, including extensive background discussions that make it well-rounded and easy to follow. In particular, the introduction provides a concise account of the evolution of numerical methods to address PDEs in physics, and how this ultimately led to the quantum-inspired methods discussed here.

  3. I have appreciated the wide selection of different numerical applications.

Weaknesses

However, the article also has some major issues, which I will comment on in detail in the corresponding section below. This is a broad summary:

  1. For all calculations, it is unclear which exact computational parameters with respect to the QTT representations have been employed. Judging from Fig. 10 and the discussion in Sec. 7, it seems that the maximum number of grid points the authors have investigated is 2^12, a resolution which is very much doable with “traditional methods”. As the main advantage of utilising QTT schemes is to go beyond numbers of grid points which are traditionally accessible (at least > 2^{20-25} in one dimension, say), I deem the results discussed in this work merely as proof-of-principle, rather than a "ground-breaking discovery" (as per the author's indication of the journal expectations).

  2. There are various statements that require clarification, a more accurate justification through numerics or further references to the literature.

  3. I am not quite convinced by some of the conclusions drawn, in particular from Figs. 6 and 9, as there are either small inconsistencies, inconclusive parameter ranges or insufficiently explained behaviours.

Report

The article “Quantics Tensor Train for solving Gross-Pitaevskii equation” presents a quantum-inspired numerical solution to the Gross-Pitaevskii equation (GPE) as a non-linear partial differential equation, based on quantics tensor trains (QTT). It focuses on two main algorithms: a ground state search via imaginary time evolution and a dynamical evolution of a given boundary condition via an RK4 algorithm. It explains how the necessary terms and operators (i.e., the different parts of the Hamiltonian, including the non-linear term) can be constructed as QTTs, highlighting in particular the inclusion of long-range interactions and multi-component Bose-Einstein condensates (BEC). The article then presents a few exemplary applications of these methods, including quasi-disordered potentials, Rydberg dressing and two-component BECs, before briefly discussing the numerical resources and giving an outlook on further research.

The present article is written in a clear and intelligible way and provides citations to relevant literature. The latter can still be expanded to be “as representative and complete as possible”. It contains a clear conclusion and offers perspectives for future work, as well as a detailed abstract and introduction. I could not find a link to the code used in this work, but the algorithms and techniques are well-described. However, it currently does not provide sufficient details, in particular about the implementation of the numerical experiments. Once computational details and a few more relevant references are added, the paper will meet the general acceptance criteria.

In the current version of the article, I am however a bit more hesitant on meeting Scipost Physics’s expectations, of which at least one is required. While the application of QTTs to solving partial differential equations with non-linearities is certainly novel, it is not the only work achieving that (see e.g. arXiv:2507.04262, arXiv:2507.04279, arXiv:2507.01149, which provide considerably more computational details and/or resolutions beyond the reach of traditional numerics). The presented results do not go beyond the current (or rather, previous) state-of-the-art, in that they remain completely accessible by traditional numerical methods. The algorithms applied (ground state search via imaginary time evolution, real-time evolution via RK4) are well-established and their application in the context of QTTs lacks the discussion of crucial computational details, making it hard to assess the power of the method. Further, not all results (in particular Figs. 6, 9) are currently well-discussed.

Before I can recommend the article for publication, I will therefore ask for major revisions according to my above comments. Once these revisions have been implemented and my questions answered (or the issues I have raised convincingly argued against), I am happy to recommend the article for publication with SciPost Physics. Otherwise, it may be more suited to a more technical sub-journal, highlighting for instance the mixture of BEC species and the long-range interaction as a main new algorithmic result.

Requested changes

  1. around Eq. 7, it could be useful to have a few references for the mixed-species GPE, or citing some experiments/numerics where this has been investigated.

  2. in line 179 when introducing the quantics prescription with the binary representation of x, I think it is somewhat misleading to let the summation over binary indices run until infinity, as this “approximates” a real function (as per your definition of f) by a rational function. I would suggest introducing quantics directly with a finite binary index, or otherwise would ask you to explain the chosen notation with an infinite sum more carefully.

  3. line 251, “three solutions …” Please provide explicit references to (i), (ii) and (iii) here.

  4. line 261, “sum of two functions…”: the statement is of course correct, but reflects simply the naive increase in bond dimension upon summation of two arbitrary MPS, and has nothing to do with their origin as functions. Maybe this could be somewhat reformulated to reflect that.

  5. line 274: “Tensor Cross Interpolation”: it could be good to cite the corresponding references [55-57] already here. Also, more references might be appropriate.

  6. line 311: this construction is not new, see for instance Phys. Rev. X 13, 021015. Please provide this or other corresponding references.

  7. line 336: “non-linearities…” I don’t understand the comment with respect to Rényi entropies and how it serves as “inspiration” (line 338). Could you clarify this?

  8. lines 352-355: compression of the QTT representing |psi|^2: The statement that the compressed bond dimension is smaller than the uncompressed one is trivially true, I do not quite understand the use of highlighting this so explicitly. Further, you refer to your numerical experiments, yet you provide no data about this statement. What bond dimensions can you achieve upon compression of |psi|^2 that maintain which fidelity with the original QTT of |psi|^2 ?

  9. line 363: “Alternative techniques might be useful”: why, what do these achieve in comparison to the previous discussion? It could be good to highlight this in one more sentence or to be a bit more explicit..

  10. line 371: The MPS construction with a separate part for the BEC species and for the spatial degrees of freedom is neat. How does the bond dimension behave in the “species part”? At the intersection between species and spatial, is the bond dimension trivial?

  11. line 475: why have you decided to implement the RK4 and not one of the other time-evolution algorithms? You could provide a few more references to the other time-evolution algorithms you named, in particular given that some of them have already been employed to solve the GPE numerically (e.g. Journal of Computational Physics, Volume 187, Issue 1, 318-342 (2003), and similar)

  12. line 507: Here is a crucial discussion of further numerical details missing. How many tensors/which spatial discretisation do you use? What are your imaginary time increments? How many evolution steps did you need to perform until you observe convergence to the ground state, with what convergence criterion? How did the bond dimension evolve during this process? How does this behave for different spatial discretisations? In its current form, this is a result with no context that does not give any information about potential advantages of the QTT approach over conventional vectorised methods.

  13. line 510: “double well potential”: I do not understand this description, given that the potential has been defined as a Gaussian \kappa exp(-x^2/2).

  14. line 513: A reference to the Thomas-Fermi profile could be helpful.

  15. line 523: “full agreement up to numerical precision with standard GP solvers”: Where is the data to support this claim? What “standard GP solvers” are you referring to?

  16. Sec. 6.2: Same as in the previous section, please provide the details of your computation.

  17. Fig. 5: Is plotting it down to 1e-15 meaningful? Judge with respect to the convergence/cutoff parameters (that are not explicitly mentioned). Panel b), the 10^2 y-label is slightly cut off.

  18. Fig. 6: The colourmap labels in panels (a, b, c, e, f), which describe the probability density, exceed 1. Is this a mistake or a normalisation issue? Please investigate and fix this. Further, the evolution time until t=5 is on the quite short side to see the emergence of quasiperiodic effects. How have you decided to evolve until this time? Have you considered larger times? Again, crucial computational details are missing. What is the size of your time steps, your spatial discretisation? How many evolution steps did you perform, how long did it take? How did the bond dimension evolve during this process, or did you impose a strict upper bond on the bond dimension?

  19. Fig. 6 and discussion, continued: the data points for g=1, Delta=200 and g=1, Delta=0 almost coincide, which is in contradiction with your statements in lines 556, 557. Why then would the line g=1, Delta=300 look so different, and seemingly follow a similar trend as the lines g=0, Delta=0, 200? Further, the statement about “repulsive interactions” (line 559) seems incorrect to me, as, judging from the plot, you are referring to the g=0 lines here. The g=-1 (the repulsive) lines all bunch up at the bottom. Overall, I am therefore not yet convinced by the conclusions drawn from the data here, and I would encourage checking longer evolution times (on the order of t=50-100, if computationally feasible); there may be large-scale oscillation effects of the width before convergence settles in.

  20. Sec. 6.3: same comment about missing computational details.

  21. line 567: is beta << 1 or is beta = 2 (as in the caption of Fig. 7)?

  22. line 577: what is the initial state of your imaginary time evolution, or how did you choose it?

  23. Sec. 6.4: same comment about missing computational details.

  24. line 600: Why did you choose a soliton solution (and not, for instance, two propagating Gaussian wave packets)? Please also provide some references for the soliton solution. I am further not quite convinced of the statement “which corresponds to two localized BEC”, as solitons rather occur in BECs (instead of describing the whole BEC)?

  25. line 604: “inside a trapping potential”: which potential do you use? Please provide the explicit functional form.

  26. line 615: “two branches of the wave function meeting at the same point”: I do not quite understand this comment, as in all of the cases shown, the propagating wave functions cross each other multiple times?

  27. line 616, “the pattern is a bit different”: Please describe this in more scientific terms.

  28. line 620: “we expect repulsion…”: ok, you expect it, but don’t observe it - could you explain this a bit more? This kind of counterintuitive behaviour should be understood better. Also, you have g >> g_ij (60 >> 1), could this be a reason that the inter-particle repulsion is simply very small compared to the intra-particle non-linearity? Actually, why do you expect repulsion? Shouldn’t this only be the case for g < 0 (both kinds of g)?

  29. line 621: “species meet briefly”: (as above) I do not understand this comment; how (and where) does this happen in panels 9(g,h), but not e.g. in 9(a,b)? What exactly does “meeting” even mean, just the intersection of the curves at specific times in the left and right panels? Maybe it could be helpful to show the evolution of both species in the same panel, maybe with different colour maps?

  30. line 623: “in full agreement”: how do you assess “full agreement”?

  31. Fig. 9: In general, I think the exemplary parameter values are not chosen very well, as only panels (e, f) are appreciably different from the remaining panels, with (a, b) and (g, h) looking almost identical (yet corresponding to vastly different non-linear regimes - why are they so similar?). As such, we can’t really identify different regimes from this figure. The interesting parameters are lambda (the inter-species tunneling) and g_ij (the inter-species non-linear interaction). It would be more meaningful to vary those separately from each other, e.g. four values, each, and then show on a 4x4 grid how that influences the propagation of the wave functions. Smaller things: x label 20 is consistently missing on the left; the caption mentions the absolute value |psi|, but I assume you mean the probability density, as shown in the colourbar label.

  32. Sec. 7: overall: you seem to check spatial discretisations until 2^12 grid points. As stated before, this is very much in reach of traditional methods, especially when using sparse matrices, thus representing a regime in which QTTs offer no effective advantage. In certain cases, such as the examples you describe in lines 643-656, the discretised wave function may be stored with fewer parameters in a QTT than in a traditional finite-difference vector, already for these relatively coarse grids, granted. Yet, the computational overhead of performing QTT manipulations will likely eat away all the advantages from a parameter reduction by a factor of roughly 2. You even conclude in lines 674-676 that “traditional methods [...] are more optimal than QTT” (using sparse matrices, in the context of applying the Hamiltonian, etc.), somewhat weakening the whole chain of your reasoning.

  33. line 653: “bond dimension required is 9”: this will depend on other parameters of your computation, such as relative per-tensor error tolerances. Please report on these and/or explain where the bond dimension of 9 comes from.

  34. line 681: “we can imagine that for a large enough number of grid points, the QTT method will be more efficient than the sparse method”: this is exactly the kind of statement you should be able to quantify, without data, this is nothing but a claim, and not a very accurate one. To make a strong statement, Fig. 10b should be pushed to a spatial discretisation where QTTs outperform also the “traditional sparse” methods.

  35. line 686: “computational gain… more visible in higher-dimensional geometries”: I agree with you in terms of the number of tensors, which you argue correctly leads to a linear increase in the parameter count compared to vectorised methods. Whether it is “more visible”, though, is a non-quantitative statement of speculation. QTTs for higher-dimensional systems in a PDE setup may exhibit a strong increase in the bond dimension, making MPO-MPS manipulations considerably more costly. While I agree this is of course still better than vectorised methods and, at the current time, the way to go forward, I would encourage you to rephrase this statement in a somewhat more nuanced and careful way, and potentially cite more references employing QTTs for higher-dimensional systems.

  36. lines 695-698: “we proposed”: no, as you state, these algorithms are very established and seem to be used without major changes to the algorithms themselves. I would encourage you to change the wording. “under a controlled error threshold”: As discussed before, you give no details about which error thresholds you have applied, how they are controlled, etc. Please fix this first before drawing these types of conclusions.

  37. line 700: “we benchmark our results with standard methods, [...] obtaining agreement on the numerical precision” Where do you do this? I see no benchmarking in this paper at all. I would encourage you to implement the algorithms you discuss in a vectorised version, at the spatial resolutions employed, those will run in seconds. From this, you can define point-by-point error metrics and monitor the precision of the QTT methods. Further, what is “agreement on the numerical precision”? Do you mean “up to numerical precision”? As of now, your statement is akin to saying “the numerical precision of both methods (QTT/traditional) agrees”, which would be a not very precise statement, as both methods will suffer from various time-evolution, discretisation errors, etc., but the QTT methods will incur additional errors due to compression, TCI errors, etc.

  38. line 707: “dimension-agnostic”: This remains to be carefully shown. While I certainly hope for this, too, this is a claim, not a conclusion. Same for the extensions to two and three dimensions.

Further, there are a few minor/stylistic issues, which I will list below:

  1. the font and font size between e.g. figure panel labels is inconsistent (see e.g. Fig. 2, 3 vs Fig. 4).

  2. there are various typos, missing articles etc. in text (e.g. line 162, stronGly-correlated, 491: sinGULAR value decomposition). While it remains a very nicely written paper, I would encourage the authors to check the spelling and language more carefully.

  3. in several lines, the text extends across the line break

  4. not all references provide a hyperlink, and not all references are in the same style (e.g. [6] vs [22], and others).

  5. inconsistent caps in titles, e.g. Secs. 3, 3.3, 3.4, 4.3 compared to others

  6. I could not find a link to the code used in this paper.

  7. line 499: What is the definition of HDAF? I think it would be better to provide it.

Further suggestions, that may also be ignored:

  1. around Eq. 11, you could choose to define the notations more carefully (normalisation of c, dimension of each local Hilbert space, e.g. s_i = 1….d_i). It is clear as is, but might fit in more nicely with these small additional details.

  2. Eq. 24: It could be helpful to underscore the left group of tensors with “species” and the right group with “space” (or similar), in analogy to Fig. 2c.

  3. Eq. 28: I’m not sure if the fractal symbol is an optimal choice of notation, given that it is not re-used in the remainder of the paper (meaning, it is a somewhat hard-to-read symbol for a quantity that may not require a separate definition).

  4. line 274: Given the explicit nature of your explanations and the nice review aspect of your paper, it could also be good to discuss TCI in a few more sentences, but this is just a suggestion.

  5. Fig. 6: I think it could be helpful to put the parameters in the respective panels, next to (a), (b) etc. Makes it much easier to recognise which panel shows what. Also for panel (j), the legend is hard to decipher, and some green tones are fairly similar (which is fine for any gradual transition, but here you have hard transitions between parameter values for g). I would suggest e.g. a colour-coding only for Delta (as this is “turned on”), and showing the different g’s e.g. via differently dashed lines.

  6. Fig. 8: It would help to show the parameter values next to (a), (b) etc. in the panels.

  7. Sec. 6.4: It could be nice to reproduce the multi-component GPE in this section, for ease of reading.

  8. Fig. 9: Same comment about adding the parameter values directly to the panels.

Recommendation

Ask for major revision

---

## Round 1 · Referee Report · Anonymous (Referee 2) · 2025-7-29

Strengths

  1. The paper introduces a cross-disciplinary use of QTT compression for nonlinear GP equations, potentially opening .

  2. The extension of QTT toward solving non-linear equations could be useful in broad ranges of systems.

  3. Manuscript is well structured, with extensive background that can help non-tensor-network readers.

Weaknesses

  1. I find that this paper falls somewhat short in terms of novelty. A substantial portion of the content consists of existing materials, and the main contribution on the methodological advancement appears primarily in Section 4, which constitutes a relatively technical improvement.

  2. The evidence provided for the claimed performance advantages is insufficient. The proposed method is only evaluated in one-dimensional, moderately sized problems, and in the test case, traditional methods using sparse matrices remain faster. It is unclear under what relevant conditions / sizes the claimed “exponential” improvement would offer a practical advantage over sparse-matrix-based GPE solvers.

  3. I find the results (figures) of the paper miss some details. Given the inherent trade-off between accuracy and computational cost in the QTT method, governed by bond dimension or truncation error, the authors should specify the bond dimensions or truncation errors used in generating the results. Furthermore, it would be helpful to indicate how close these results are to those obtained from an exact GPE solver. Similarly, in Fig. 10, it is not entirely clear under what assumptions or configurations the QTT operation counts are estimated.

As there should be a tradeoff between the accuracy and the computation cost of QTT method (controlled by the bond dimension / truncation error), the author should clearly specify what bond dimension / truncation error are used for getting the plots they showed, and how close the results are to the exact GPE solver. Similarly, it is not fully clear in Fig.10 that under what condition that QTT number of operations is estimated.

I also suggest that the authors include an analysis showing how the bond dimension cutoff or truncation error affects the GPE simulation results. For instance, even if low bond dimensions do not reproduce the exact results, do they still provide a qualitatively accurate solution to the GPE?

  1. Some side comments: (1) Since this method aim to provide a GPE solver, it may attract interest from researchers who wish to apply it in practice. It would therefore be beneficial (though not required) if the authors consider open-sourcing their code, as this could help the community engage more readily with their work.

(2) I also noticed some inconsistent or improper use of notation, such as applying big-O notation to numerical values. A check of the mathematical notation throughout the paper would help improve clarity and rigor.

Report

I think this paper has good potential to provide a new approximate solver for GPE, which could open a new pathway in the classical simulation of BEC systems. Although due to my concern above, I recommend that the authors revise the manuscript before it can be considered for publication.

Recommendation

Ask for major revision

---

## Round 1 · Referee Report · Anonymous (Referee 3) · 2025-8-10

Strengths

  • Topic of great interest to the community
  • Manuscript is well written and provides a detailed introduction

Weaknesses

  • Quasi-disorder results should be clarified
  • Methodology is not shown to surpass a conventional method

Report

The authors present an algorithm based on tensor networks to solve the Gross-Pitaevskii equation. Their methodology is based on tensor trains, which allows them to use tensor network algorithms to perform real and imaginary time evolution. The application of tensor network algorithms to effectively classical problems is a rising new direction in computational physics, and may enable tackling exponentially hard classical problems using quantum inspired algorithms. This is an exciting direction in computational physics, and the authors make a step in this direction by tackling a non-linear time-dependent differential equation. While the topic on its own is very interesting, I believe that the work of the authors needs to be revised, including performing some further analysis, before it can be accepted in Scipost Physics. I write my specific comments below.

Requested changes

1 - The authors refer to “quasi-disorder” and “disorder” potential. I believe that the system would be better characterized as quasiperiodic, and in the instance considered by the authors as periodic. Specifically, in Eqs. 42, 43 it is not clear what is incommensurate if one takes that the discretization dx is infinitesimally small. In my understanding the authors are targetting the continuum limit, and in that regime rational and irrational frequencies of the modulation are equivalent, given that the problem does not have another length scale. In case they are not equivalent, the authors should show how their “quasi-disorder” is qualitatively different from just a potential with rational frequency. Overall, I would remove reference to this potential as “disorder”, as in my understanding, in the limit the authors consider, their problem is genuinely different from a disordered one.
2 - In relation to their benchmark, currently the calculations of the authors are below what can be done with traditional sparse methods. I believe that the authors should show that their methodology allows reaching some regime unreachable for conventional methods, or that a specific problem can be solved much faster with their algorithm than with a traditional sparse methodology. Of course, it is not necessary that this methodology is better than a sparse one in all instances the authors study. However, I believe that in order to demonstrate the usefulness of this tensor network method, a clear advantage in at least one case must be shown.
3 - This is a methodology manuscript, and therefore I believe that paper would benefit from a more detailed analysis of the error in their algorithm, focused on one of the specific problems that the authors study. Specifically, the authors can address the errors with respect to the bond dimension and length of the time evolution. This will be of special interest to the specialized community, as it will highlight how the errors scale with the bond dimension for the algorithm the authors present.
4 - I would strongly encourage the authors to share online the code that they developed implementing their algorithm, and that they used to perform the calculations of their manuscript. I believe that this would be of great interest to the community, and would substantially increase the impact of their manuscript. The authors could upload the code to a repository such as Github/Gitlab/Zenodo, and include a statement in their manuscript as “The code and algorithms to reproduce our results can be bounds in [address]”

To summarize, I believe that the current paper explores a very interesting idea in the field, and it can be of interest to the specialized community. However, I firmly believe that the results presented in their manuscript have to be extended and improved before the paper can be published in Scipost Physics.

Recommendation

Ask for major revision

---

## Editorial Decision

awaiting_resubmission